# ADAPTIVERESIDUAL: INFERENCE-TIME TRUST CALIBRATION FOR CONTEXTUAL KNOWLEDGE INJECTION

## ABSTRACT

Modern large language models (LLMs) commonly leverage external context (*e.g.* Retrieval-Augmented Generation, RAG) to provide more accurate and up-to-date information. However, recent research reveals that once conflicts arise between the contextual information and the internal parametric knowledge, LLMs tend to underutilize the external evidence, leading to unreliable or even contradictory outputs. This raises a fundamental question: *how can we dynamically reconcile these knowledge conflicts to ensure faithful integration of contextual information*? Inspired by mechanism interpretability findings that identify the `Attention` module as the primary aggregator of external context and the `FFN` module as the locus of internal knowledge lookup, we pinpoint the vanilla residual pathway as the crucial junction where these two information streams are integrated. Based on this insight, we introduce AdaRes (Adaptive Residual), a lightweight, parameter-free trust calibration mechanism that operates at test-time. Specifically, AdaRes recalibrates the standard residual connection to dynamically balance the influence of external knowledge (from `Attention`) and internal knowledge (from `FFN`). This balancing is guided by two instance-specific "trust scores", which are calculated on-the-fly by probing how much the input query relies on contextual versus parametric knowledge sources. By adaptively reweighting these contributions without altering any parameters, AdaRes effectively mitigates knowledge conflicts. Experiments on different benchmarks verify the effectiveness of AdaRes in regulating contextual and parametric knowledge.

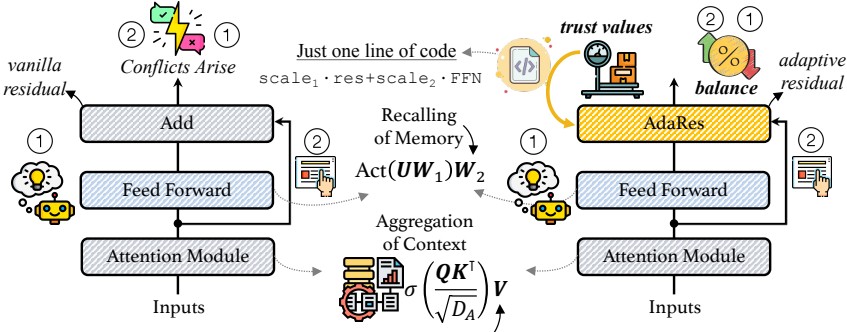

Figure 1: Model outputs become unreliable when conflicts arise. The vanilla residual connection merely adds knowledge from two different sources *equally* (*i.e.* the contextual and parametric), which cannot handle conflicts. *Our motivation is to introduce trust values to enable reconciliation.*

## 1 INTRODUCTION

Large language models (LLMs) have demonstrated remarkable performance due to encoding broad parametric knowledge and exhibiting strong in-context learning capabilities, thereby supporting applications ranging from question answering to knowledge-intensive dialogue (Dai, 2024; Wu et al., 2024a; Hu et al., 2025). In practical applications, tasks often require information that lies outside the

parametric memory, including recent facts, domain knowledge, retrieved passages, or user-provided context, so that LLMs' predictions could align with real requirements. A prevailing strategy supplies such information directly in the input context, *e.g.* via retrieval-augmented generation (RAG), a practice typically referred to as contextual knowledge injection (Yang et al., 2023; Ovadia et al., 2024; Wang et al., 2025a; Bhushan et al., 2025). While effective, this paradigm exposes a fundamental vulnerability: when the provided context conflicts with the model's parametric knowledge, LLMs often ignore the new evidence, even leading to factual errors or unreliable behavior (Xu et al., 2024b;a; Xie et al., 2024). This bias limits the effectiveness of applications such as RAG systems and fact correction, all of which rely on the model's ability to incorporate external knowledge (He et al., 2024; Wang et al., 2025b). Existing remedies either permanently modify model parameters (too static) or passively prepending context (fails to resolve conflicts), neither of which provides **a dynamic and principled paradigm for reconciling knowledge conflicts at inference time**.

Inspired by LLMs' studies (*i.e.* identifying `FFN` (Feed-Forward Network) as primary repositories of parametric memory (Geva et al., 2021; 2023; Hernandez et al., 2024; Zhai et al., 2025b), with `Attention` acting primarily to aggregate contextual information (Wang et al., 2023; Lu et al., 2024; Chen et al., 2024), see Appendix C for explanation), we revisit the residual connection in each LLM hidden layer and discover that it provides an intrinsic vantage point for mediating the influence between parametric knowledge and contextual information (see Figure 1). Recognizing this viewpoint, we introduce *adaptive residual* (dubbed AdaRes), a lightweight, parameter-free mechanism that dynamically reconciles the contributions of internal knowledge (from `FFN`) and external information (from `Attention`) within the residual pathway. This process is governed by two trust values that quantify the contributions of these knowledge sources toward resolving the input query.

Building on this view, we estimate, at the layer level, the trust in the context and the parametric knowledge for the given query without updating any parameters. Specifically, the context trust is obtained by a probe within `Attention` that measures the query's reliance on the context. Complementarily, the trust value for parametric knowledge is produced via an analogous probe in `FFN` that scores affinity to the feed-forward memory. Given these trust values, we employ a *Lambda Function* to perform the reweighting process. Additionally, AdaRes is applied at test time to a chosen subset of hidden layers and provides explicit control over the relative influence of two knowledge sources. Extensive experiments on four benchmarks verify the effectiveness of our adaptive residual. To summarize, our contribution could be listed as follows:

- **A Novel Reconciliation Mechanism:** We propose AdaRes, a parameter-free modification to the residual pathway that dynamically mediates between contextual and parametric knowledge. It is model-agnostic and can be integrated into existing LLMs with minimal code changes.

- **Train-Free Trust Calibration:** We design intuitive, training-free methods to estimate the trust in contextual and parametric knowledge and provide a *Lambda Function* to utilize this trust.

- **Empirical Verification:** Extensive experiments on conflict-centric benchmarks verify the effectiveness of AdaRes in conflict reconciliation, with negligible runtime cost.

## 2 RELATED WORK

### 2.1 ANALYSIS: LLMS' BEHAVIOR UNDER CONFLICTS

Contextual knowledge injection often materializes as retrieval-augmented generation, which conditions generation on retrieved information (Wu et al., 2024b; He et al., 2024; Wang et al., 2025b). Understanding how LLMs behave when discrepancies between contextual and parametric knowledge arise is crucial, since unpredictable outcomes undermine reliability. Tan et al. (2024) examine open-domain QA and show that models often favor parametric knowledge when retrieval is incomplete. In contrast, Xie et al. (2024) construct controlled conflicts between memorized facts and curated external context and show high receptivity to coherent, persuasive external evidence even against parametric beliefs. Complementarily, Qian et al. (2024) find frequent deviations from parametric recall under conflicts. Farahani & Johansson (2024) indicate that, when both sources are available, reliance often shifts toward the context. Under interactive and multi-turn settings, Xu et al. (2024a) report a preference for logically structured presentations even when such structure conflicts with factual accuracy. Taken together, **what is the conclusion? No universal rule for whether an LLM prioritizes contextual or parametric knowledge** (Xu et al., 2024b).

## 2.2 Solutions: Handling Conflicts

To reconcile contextual and parametric knowledge, prior works can be organized into four categories (Xu et al., 2024b). **Context-first** alignment methods prioritize generation toward supplied context, such as preserving factual consistency via prompting strategies (Zhou et al., 2023); enhancing situated faithfulness (Huang et al., 2025; Zhang et al., 2025a;b), amplifying distributional differences between decoding with and without context (Shi et al., 2024); intervening on internal model components (Jin et al., 2024b; Li et al., 2025); or knowledge editing (Wang et al., 2024a; Xu et al., 2025; Zhai et al., 2025a; Fang et al., 2025). **Parametric-preservation** strategies safeguard internal memory when conflicts arise. Some works reduce contextual noise or verify parametric beliefs before answering (Pan et al., 2023), or instruct models to verify memorized knowledge (Xu et al., 2024a), or adjust influential attention heads to downweight low-quality retrieved content (Deng et al., 2025). **Dual-response strategies** first detect conflicts, and then invoke tailored resolution strategies, such the design of a three-step identifying process (Wang et al., 2024b). **Fusion** methods merge information from both sources. Zhang et al. (2023) use discriminators to pair compatible generated and retrieved passages for joint use. Jin et al. (2024a) adopt contrastive decoding to maximize differences of logits under conflicts.

**Positioning This Work** Existing studies provide limited control over the relative impact of contextual aggregation versus parametric recall inside LLMs. Our work differs by introducing a lightweight *adaptive residual* mechanism that merely augments the standard residual pathway with the calibrated trust to reweight contributions from different knowledge sources explicitly.

## 3 Methodology

### 3.1 Notation and Task Definition

Let $X = [C; Z]$ denote the string-form *context-query* input, where $C$ and $Z$ are *contextual* knowledge and *query* prompt, respectively. $[;]$ denotes concatenation. This textual input is first tokenized into token sequences $X = [x_1, \ldots, x_{|X|}]$ with $|X|$ being the sequence length, and then these tokens are encoded by an LLM into hidden representations. We employ the **bold** uppercase $\boldsymbol{X}^{(l)}$ to denote the whole token feature matrix at $l$-th hidden layer, with the **bold** lowercase $[\boldsymbol{x}_1^{(l)}, \ldots, \boldsymbol{x}_{|X|}^{(l)}]$ being token features. $l \in \{1, ..., L\}$ indexes model layers. The LLM is $\mathcal{M}_{\boldsymbol{\Theta}}$, with fixed parameters $\boldsymbol{\Theta}$.

To analyze potential conflicts within a given model, we first detail each LLM block as:

$$\boldsymbol{U}^{(l)} = \underbrace{\texttt{Attention}(\texttt{Norm}(\boldsymbol{X}^{(l)}))}_{\textit{Aggregating contextual information: } \boldsymbol{A}^{(l)}} + \boldsymbol{X}^{(l)}; \quad \boldsymbol{X}^{(l+1)} = \underbrace{\texttt{FFN}(\texttt{Norm}(\boldsymbol{U}^{(l)}))}_{\textit{Recalling parametric knowledge: } \boldsymbol{F}^{(l)}} + \boldsymbol{U}^{(l)} \quad (1)$$

Here, the prevalent Pre-Normalization setting is used, *i.e.* $\texttt{Norm}(\cdot)$. $\boldsymbol{X}^{(l)}$ and $\boldsymbol{X}^{(l+1)}$ refer to the input and output representations of $l$-th layer, respectively. $\texttt{Attention}(\cdot)$ is the attention module that aggregates *contextual* information while $\texttt{FFN}$ performs *parametric* knowledge recall. Conceptually, the information flow within this layer can be simplified as follows:

$$\boldsymbol{X}^{(l+1)} = 1 \cdot \boldsymbol{A}^{(l)} + 1 \cdot \boldsymbol{F}^{(l)} + \boldsymbol{X}^{(l)} \quad (2)$$

This formulation reveals that the vanilla residual pathway passively aggregates two different sources of knowledge, *i.e.* $1 \cdot \boldsymbol{A}^{(l)} + 1 \cdot \boldsymbol{F}^{(l)}$. It lacks any mechanism to selectively prioritize or reconcile these information streams, leading to unpredictable behavior when they conflict.

### 3.2 Adaptive Residual

To make up for the deficiency of Eq. 2 in reconciling $\boldsymbol{A}^{(l)}$ and $\boldsymbol{F}^{(l)}$, we introduce the *adaptive residual* (AdaRes for short) to reweight the two terms while leaving parameters unchanged:

$$\boldsymbol{X}^{(l+1)} = \Lambda(\boldsymbol{A}^{(l)}, \boldsymbol{F}^{(l)}; \alpha^{(l)}, \beta^{(l)} | \mathcal{H}) + \boldsymbol{X}^{(l)} \quad (3)$$

where $\Lambda$ is the *Lambda Function* to scale the features from different sources. In addition to the features $\boldsymbol{A}^{(l)}$ and $\boldsymbol{F}^{(l)}$, $\Lambda$ also receives trust values $\alpha^{(l)}$ and $\beta^{(l)}$ to compute the scale factors, *i.e.* $\text{scale}_1$ for $\boldsymbol{A}^{(l)}$ and $\text{scale}_2$ for $\boldsymbol{F}^{(l)}$. $\mathcal{H} \subseteq \{1, \ldots, L\}$ refers to the set of hidden layers where the *Lambda*

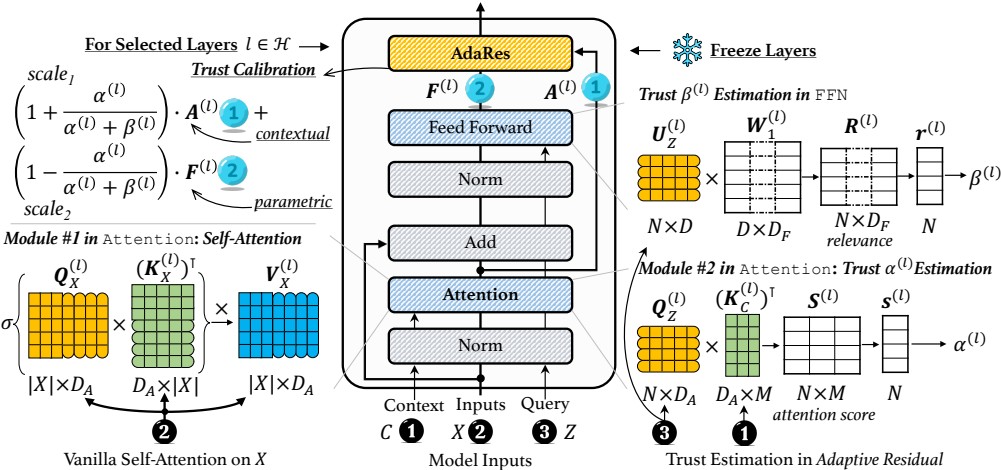

Figure 2: Framework of the proposed AdaRes. Specifically, residual pathways for *context informa-tion* $C$ and *query prompt* $Z$ are omitted to enhance simplicity. All the model parameters are frozen.

*Function* $\Lambda$ is applied. For any layer $l \notin \mathcal{H}$, it will seamlessly revert to the vanilla form in Eq. 2. Fig-ure 2 depicts the layer equipped with our method, where the AdaRes layer differs from the original only in the recalibrated residual connection. Our central idea is to use this pathway to dynamically balance the contributions from contextual and parametric knowledge. Consequently, two additional input streams are introduced for trust estimation, *i.e* $C$ and $Z$. Based on this depiction and definition in Eq. 3, two core steps need to be determined: *how to estimate the trust values* $(\alpha^{(l)}, \beta^{(l)})$? and *how to instantiate the function* $\Lambda$ *to reweight* $A^{(l)}$ *and* $F^{(l)}$ *based on the estimated trust?*

### 3.2.1 TRUST ESTIMATION

The trust values $(\alpha^{(l)}, \beta^{(l)})$ quantify, at the $l$-th layer, the relevance or usefulness of the (*contextual, parametric*) knowledge for answering the current query. Taking $\alpha^{(l)}$ as an example, we estimate it from the input *context-query* pair $(C, Z)$. To precisely measure the query's reliance on the context, we process $C$ and $Z$ through the attention mechanism *separately* to generate their respective feature matrices, thus avoiding information leakage from the standard self-attention. This is also why we introduce *three* input information streams in Figure 2. Specifically, the core idea of estimation is to measure the extent to which the *query* $Z$ attends to the *context* $C$ via a cross-attention probe:

$$\boldsymbol{S}^{(l)} = \sum_{\text{Head}} \sigma\left(\frac{\boldsymbol{Q}_Z^{(l)}(\boldsymbol{K}_C^{(l)})^\top}{\sqrt{D_A}}\right), \quad \boldsymbol{S}^{(l)} \in \mathbb{R}^{N \times M}; \quad \boldsymbol{s}^{(l)} = \frac{1}{M}\boldsymbol{S}^{(l)}\mathbf{1}_M, \quad \boldsymbol{s}^{(l)} \in \mathbb{R}^N; \quad (4)$$

Here, $\boldsymbol{S}^{(l)}$ is the *query*-to-*context* attention score that reflects the potential usefulness of the contex-tual knowledge; $\boldsymbol{s}^{(l)}$ averages its attention mass for each query token over the context span. $N$ and $M$ are the lengths of token sequences for *query* $Z$ and *context* $C$. $\boldsymbol{Q}_Z^{(l)}$ and $\boldsymbol{K}_C^{(l)}$ denote the feature matrices in one attention head with $D_A$ being its dimension. $\sum_{\text{Head}}$ is summation of different at-tention heads. $\sigma$ denotes the softmax function. $\mathbf{1}_M$ is an all-ones column vector of length $M$. From this per-token score vector $\boldsymbol{s}^{(l)}$, we estimate $\alpha^{(l)}$ by taking its mean, *i.e.* $\alpha^{(l)} = \text{Mean}(\boldsymbol{s}^{(l)})$. This scalar value represents the expected attention from the query $Z$ to the context $C$, providing a holis-tic measure of the context's overall relevance. Analogously, $\beta^{(l)}$ is obtained by probing the FFN module with the query features (see Appendix D). Algorithm of AdaRes is provided in Appendix E.

### 3.2.2 LAMBDA FUNCTION: $\Lambda$

Given the estimated trust $(\alpha^{(l)}, \beta^{(l)})$, the function $\Lambda$'s role is to adaptively combine the contextual information $A^{(l)}$ and the parametric knowledge $F^{(l)}$. To motivate our design, we first consider the four archetypal scenarios an LLM might face, as depicted in Figure 3. While synergistic integra-tion (**Scenario #1**) is interesting (see Appendix F), this work primarily focuses on resolving the

knowledge conflicts inherent in **Scenario #4** (faithful contextual information injection). In this explicit conflict scenario, LLMs must prioritize using context information. We design a re-weighting strategy to intelligently amplify the trusted source while suppressing the other:

$$\Lambda(\boldsymbol{A}^{(l)}, \boldsymbol{F}^{(l)}; \alpha^{(l)}, \beta^{(l)} | \mathcal{H}) = (1 + \frac{\alpha^{(l)}}{\alpha^{(l)} + \beta^{(l)}}) \boldsymbol{A}^{(l)} + (1 - \frac{\alpha^{(l)}}{\alpha^{(l)} + \beta^{(l)}}) \boldsymbol{F}^{(l)}, \quad l \in \mathcal{H} \quad (5)$$

This asymmetric scaling design has a crucial property[1]: for the prioritized contextual source, its trust value is bounded in the range $[1, 2]$, while the suppressed source is bounded in $[0, 1]$. This acts as a *protective floor*, guaranteeing that the prioritized information is at least passed through (coefficient of 1, the worst case) and can be amplified up to a factor of 2 (the best case).

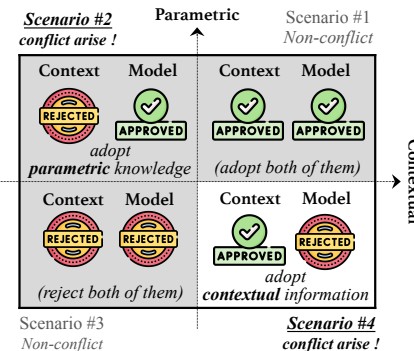

Figure 3: Which knowledge to use?

Apart from the trust estimation and the *Lambda Function*, the last key design choice is the selection of target layers $\mathcal{H}$. This is the sole hyperparameter in AdaRes, indicating that AdaRes incurs minimal tuning overhead, facilitating efficient deployment (see detailed discussion in Section 4.4). Additionally, our overall design philosophy prioritizes simplicity. Hence, two primary components (*i.e.* trust estimation via attention probing and linear scaling in $\Lambda$) are intentionally concise. Practitioners can readily substitute more advanced, task-tailored estimators or scaling functions as needed.

## 4 EXPERIMENTS

### 4.1 EXPERIMENTAL SETUP

Our experiments are designed to evaluate AdaRes's ability to reconcile knowledge conflicts, thereby enhancing the integration of contextual information in LLMs. We assess this capability on two primary classes of benchmarks: knowledge editing datasets (including ZsRE and COUNTERFACT) and conflict-aware QA datasets (*i.e.* ConflictQA (Xie et al., 2024)). The knowledge editing task is particularly well-suited to our evaluation, as it requires the model to override its internal, often conflicting, knowledge and faithfully adhere to the new fact provided in the context. All datasets are publicly available[2], and detailed experimental configurations are provided in Appendix J. All experiments of all methods are conducted on a *Linux NVIDIA A100 80GB* (256 GB RAM) machine.

### 4.2 PERFORMANCE COMPARISON: PRIORITIZING CONTEXTUAL INFORMATION

Table 1 presents the primary results on ZsRE and COUNTERFACT using three standard metrics (see Appendix J.2): *Efficacy* (adherence to the contextual fact), *Generality* (generalization to paraphrased queries under providing the same contextual information), and *Locality* (retention of unrelated parametric knowledge). The "*Original*" rows quantify the base models' pre-existing knowledge, revealing the severity of the conflict. For instance, on COUNTERFACT, Llama3 (8B)'s *Efficacy* of $0.87\%$ indicates that over $99\%$ of facts in this dataset directly conflict with its parametric memory. Generally, We compare AdaRes against a strong suite of specialized baselines in this task, including finetuning (FT-C, LoRA), parameter-editing (*e.g.* ROME, AlphaEdit), and memory-based methods (GRACE, WISE). The results indicate that extensive existing methods struggle to resolve these conflicts, with some even yielding counter-productive results. Notably, the standard In-Context Knowledge Editing (IKE) provides essential knowledge into the context, yet exhibits poor fidelity (*e.g.* Llama3 (8B) achieves $0.55\%$ *Efficacy* on COUNTERFACT with IKE). This failure underscores the critical need for an explicit mechanism to make models trust external evidence. In stark contrast, our AdaRes improves the model's adherence to the context, boosting the *Efficacy* of Llama3 (8B) from $0.55\%$ to $65.45\%$ on COUNTERFACT, a more than 100-fold increase. Our analysis reveals two

---

[1]See Appendix G for a detailed discussion on why we design $\Lambda$ in this way.

[2]Code, datasets and running scripts are all provided as supplementary materials to facilitate the reproduction.

Table 1: Performance comparison for **Scenario #4** under the consecutive knowledge editing.

| Method | CounterFact | | | ZsRE | | |
|---|---|---|---|---|---|---|
| | *Efficacy* | *Generality* | *Locality* | *Efficacy* | *Generality* | *Locality* |
| **Llama3** (8B) (Original Model) | 0.0087 | 0.0075 | / | 0.2627 | 0.2598 | / |
| FT-C (Meng et al., 2022) | 0.0575 | 0.0047 | 0.0013 | 0.0769 | 0.0666 | 0.0069 |
| LoRA Xu et al. (2024c) | 0.0077 | 0.0117 | 0.0017 | 0.1145 | 0.1116 | 0.0535 |
| ROME (Meng et al., 2022) | 0.2507 | 0.1323 | 0.0097 | 0.0339 | 0.0280 | 0.0015 |
| R-ROME (Gupta et al., 2024) | 0.4892 | 0.3662 | 0.0147 | 0.0271 | 0.0243 | 0.0035 |
| MEMIT (Meng et al., 2023) | 0.0000 | 0.0000 | 0.0722 | 0.0000 | 0.0000 | 0.0396 |
| AlphaEdit (Fang et al., 2025) | 0.0033 | 0.0017 | 0.0007 | 0.0001 | 0.0000 | 0.0003 |
| GRACE (Hartvigsen et al., 2023) | 0.0003 | 0.0000 | 0.9938 | 0.0624 | 0.0095 | 1.0000 |
| WISE (Wang et al., 2024a) | 0.1473 | 0.0763 | 0.9907 | 0.3348 | 0.3283 | 0.9997 |
| IKE (Zheng et al., 2023) | 0.0055 | 0.0043 | 0.6509 | 0.5233 | 0.5231 | 0.5289 |
| **AdaRes/RAG** | **0.6485** | **0.2963** | **1.0000** | **0.5912** | **0.5629** | **1.0000** |
| **AdaRes** | **0.6545** | **0.2987** | **1.0000** | **0.6571** | **0.6263** | **1.0000** |
| **Qwen2.5** (7B) (Original Model) | 0.0078 | 0.1340 | / | 0.2016 | 0.1942 | / |
| FT-C (Meng et al., 2022) | 0.0407 | 0.0150 | 0.0003 | 0.0239 | 0.0199 | 0.0029 |
| LoRA (Xu et al., 2024c) | 0.0200 | 0.0200 | 0.0020 | 0.0527 | 0.0516 | 0.0072 |
| ROME (Meng et al., 2022) | 0.0000 | 0.0000 | 0.0242 | 0.2809 | 0.2538 | 0.0924 |
| R-ROME (Gupta et al., 2024) | 0.5778 | 0.1742 | 0.4585 | 0.5910 | 0.5016 | 0.3871 |
| MEMIT (Meng et al., 2023) | 0.0000 | 0.0000 | 0.0000 | 0.0000 | 0.0000 | 0.0000 |
| AlphaEdit (Fang et al., 2025) | 0.0000 | 0.0000 | 0.0000 | 0.0001 | 0.0001 | 0.0002 |
| GRACE (Hartvigsen et al., 2023) | 0.0012 | 0.0000 | 0.9939 | 0.2905 | 0.0095 | 1.0000 |
| WISE (Wang et al., 2024a) | 0.0352 | 0.0283 | 0.0645 | 0.2747 | 0.2690 | 0.0985 |
| IKE (Zheng et al., 2023) | 0.5001 | 0.2272 | 0.5686 | 0.6974 | 0.6889 | 0.5029 |
| **AdaRes/RAG** | **0.6408** | **0.4772** | **1.0000** | **0.8374** | **0.8000** | **1.0000** |
| **AdaRes** | **0.6475** | **0.4812** | **1.0000** | **0.9264** | **0.9156** | **1.0000** |

further key insights: **(1)** Many knowledge editing methods appear less effective in editing, sometimes paradoxically exacerbating conflicts. **(2)** While AdaRes provides a substantial improvement, a performance gap to $100\%$ remains. We therefore conduct an error analysis in Section 4.6.

## 4.3 GENERALIZATION ON OTHER DATASETS & LLMS

To assess the generalization of AdaRes, we extend our evaluation to a wider range of datasets and a diverse suite of LLMs at different sizes, with results presented in Table 2. The findings consistently show that while baseline models suffer from significant knowledge conflicts and exhibit low utilization of contextual information, AdaRes provides a robust and significant performance uplift across all tested model families and sizes. Interestingly, we observe that the performance gains from AdaRes are often more pronounced on larger-scale models. We hypothesize that this is because larger models have stronger capabilities of language understanding, which provides a more impactful and necessary trust calibration, discussed in Section 4.6.

## 4.4 ANALYSIS OF $\mathcal{H}$

The selection of the layer set $\mathcal{H}$ is a critical hyperparameter that dictates where AdaRes is applied. To understand its effects, we first conduct a single-layer ablation study, applying AdaRes to only one layer at a time (results in Figure 4a and 4b). This analysis reveals that a model's sensitivity to intervention varies dramatically across its depth, with certain layers acting as "hotspots" where recalibration is more effective. Interestingly, we observe that different models exhibit distinct sensitivity patterns. For instance, Llama3 (8B)'s influential layers are distributed relatively uniformly, while Qwen2.5 (7B) tends to exhibit a bimodal mode (lower and higher layers show better effects).

Table 2: Generalization of AdaRes on other LLMs with different sizes (*Efficacy*).

| Versions | Phi3 | | Gemma3 | | Phi3 | | Gemma3 | |
|---|---|---|---|---|---|---|---|---|
| Size | (3.8B) | (14B) | (4B) | (12B) | (3.8B) | (14B) | (4B) | (12B) |
| | ZsRE | | | | COUNTERFACT | | | |
| Original | 0.3058 | 0.3298 | 0.2237 | 0.2225 | 0.1536 | 0.1512 | 0.0043 | 0.0055 |
| Vanilla Res | 0.5954 | 0.5695 | 0.4942 | 0.4734 | 0.6354 | 0.5186 | 0.6092 | 0.5417 |
| **AdaRes** | **0.8924** | **0.9301** | **0.9811** | **0.9868** | **0.8165** | **0.9076** | **0.9923** | **0.9992** |
| | ConflictQA-(PopQA) | | | | ConflictQA-(StrategyQA) | | | |
| Original | 0.6416 | 0.6663 | 0.5562 | 0.6260 | 0.7640 | 0.7769 | 0.6852 | 0.6689 |
| Vanilla Res | 0.8248 | 0.8003 | 0.7523 | 0.7702 | 0.8137 | 0.8191 | 0.7718 | 0.7710 |
| **AdaRes** | **0.8284** | **0.8358** | **0.7893** | **0.7949** | **0.8577** | **0.8699** | **0.8357** | **0.8261** |

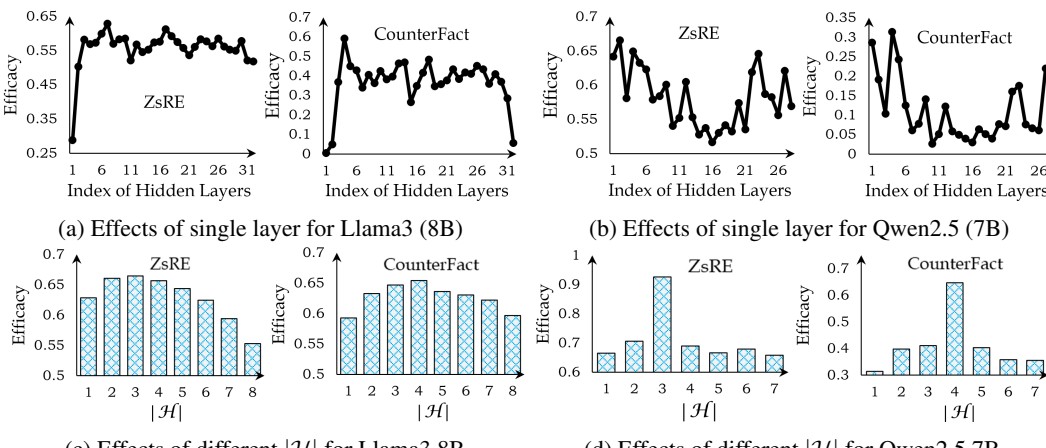

(a) Effects of single layer for Llama3 (8B)  (b) Effects of single layer for Qwen2.5 (7B)

(c) Effects of different $|\mathcal{H}|$ for Llama3 8B.  (d) Effects of different $|\mathcal{H}|$ for Qwen2.5 7B

Figure 4: Effects of $\mathcal{H}$ on model performance under prioritizing contextual information.

Informed by these identified single-layer "hotspots", we then seek an optimal multi-layer configuration. To discover an effective layer combination without an exhaustive search, we employ a *greedy forward selection* strategy. That is, starting with the best-performing single layer, we iteratively add the layer from the remaining set that yields the largest marginal performance gain, with results shown in Figure 4c and Figure 4d. The results consistently show that performance initially rises with the number of modulated layers, but peaks before gradually declining. This degradation with excessive intervention suggests a risk of "*over-correction*". Furthermore, we observe a fascinating pattern in the composition of the optimal layer sets: they consistently exhibit a *bimodal distribution*, typically combining *a cluster of early-to-mid layers* with *a cluster of higher layers* (*e.g.* $\{3, 12; 17, 25\}$ for Llama3 (8B) or $\{3; 26, 27\}$ for Qwen2.5 (7B) on COUNTERFACT). This empirical finding aligns remarkably well with established mechanistic interpretability research (Chen & Yan, 2024; Zhao et al., 2024): early-to-mid layers are often implicated in factual recall and knowledge retrieval from parametric memory, while the mid-to-high layers are responsible for high-level semantic synthesis. These results suggest a powerful takeaway for practical applications: **to make a model trust external knowledge, it is often best to intervene at both stages: first, to modulate the initial retrieval of conflicting internal facts, and second, to guide the final synthesis process to ensure the contextual information is correctly prioritized.**

### 4.5 ANALYSIS OF TRUST VALUE OF $\alpha^{(l)}$ AND SCALE FACTORS

To understand how AdaRes operates internally, we visualize the estimated trust precursors $\alpha^{(l)}$ and $\beta^{(l)}$, and the final recalibrated trust scores applied by the $\Lambda$ function. As illustrated in Figure 5, when provided with the necessary context, we observe a significant disparity between the estimated $\alpha^{(l)}$

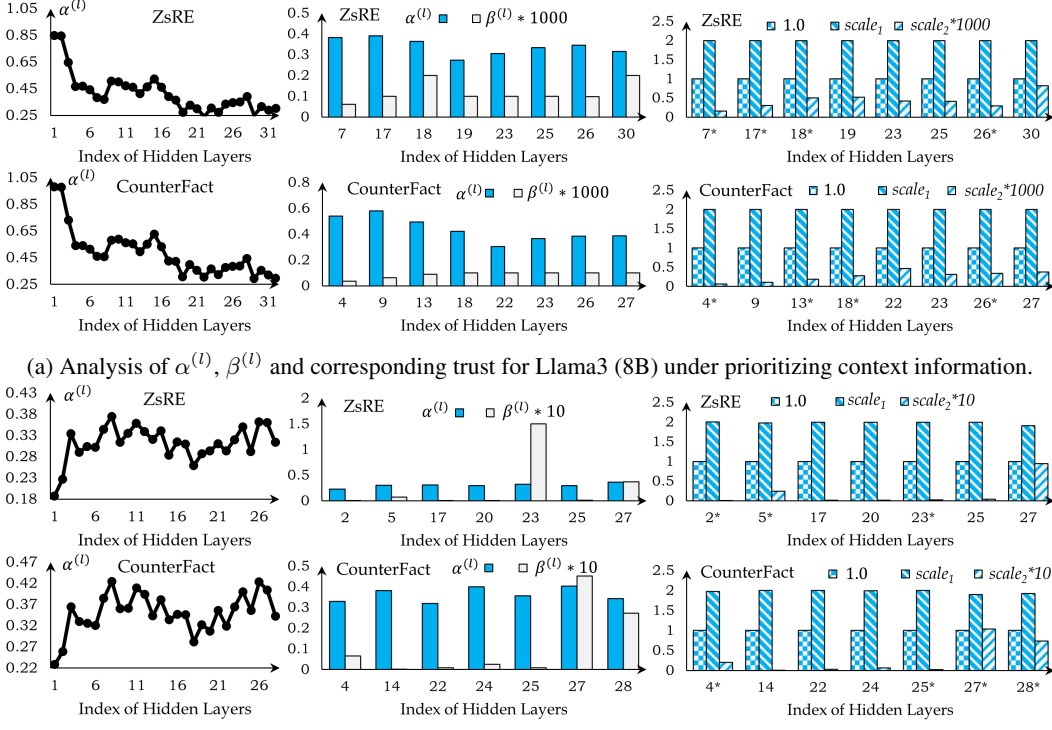

(a) Analysis of $\alpha^{(l)}$, $\beta^{(l)}$ and corresponding trust for Llama3 (8B) under prioritizing context information.

(b) Analysis of $\alpha^{(l)}$, $\beta^{(l)}$ and corresponding trust for Qwen2.5 (7B) under prioritizing context information.

Figure 5: Trend of $\alpha^{(l)}$ across layers (left). Estimated $\alpha^{(l)}$ and $\beta^{(l)}$ (middle), Adjusted trust (right).

and $\beta^{(l)}$ values across all model layers (shown in the middle of Figure 5). This indicates that our probing mechanism correctly perceives the high utility of the contextual information for the given query. However, the static, uniform weighting in the vanilla residual block ($1 \cdot \boldsymbol{A}^{(l)} + 1 \cdot \boldsymbol{F}^{(l)}$) is oblivious to this crucial signal, forcing an equal combination of a useful source with a conflicting one and thus leading to suboptimal performance. In contrast, AdaRes uses this disparity to compute highly asymmetric recalibration weights ($\text{scale}_1 \cdot \boldsymbol{A}^{(l)} + \text{scale}_2 \cdot \boldsymbol{F}^{(l)}$), which amplify the contextual information while suppressing the conflicting parametric one (shown in the right of Figure 5). Beyond these aggregate behaviors, we examine the dynamics of $\alpha^{(l)}$ across model layers, which is helpful for diagnosing the failure mode in vanilla residual. As shown in Figure 5 (left), $\alpha^{(l)}$ scores for Llama3 (8B) tends to decrease in later layers, suggesting a failure in **late-stage semantic integration**, *i.e.* the model understands the context only early on but fails to let it guide the final output. Conversely, Qwen2.5 (7B)'s trust of $\alpha^{(l)}$ are low in the initial layers, suggesting a failure in **early-stage contextual grounding**; *i.e.* it prematurely commits to its internal knowledge, and this erroneous signal propagates, making it difficult for later layers to correct the course. These distinct failure patterns provide a compelling explanation for why the bimodal intervention strategy identified in Section 4.4 is so effective. Llama3 (8B) requires late-layer intervention to force semantic adherence, while Qwen2.5 (7B) needs early-layer intervention to ensure the context is properly encoded from the start. A "*bimodal adjustment*" strategy effectively addresses both types of failure points simultaneously, providing a robust solution for different model architectures.

## 4.6 Error Analysis

As demonstrated in Table 1, while AdaRes yields substantial improvements over baseline methods, a performance gap to the ideal $100\%$ *Efficacy* remains. For instance, on COUNTERFACT, AdaRes boosts the *Efficacy* of Llama3 (8B) by $60^{+}$ percentage points compared to vanilla IKE, yet the score of $65\%$ indicates that the model does not perfectly utilize the provided context. To diagnose this, we re-think the two core components of our method: *trust estimation* and *trust utilization* (*i.e.* the $\Lambda$ function). We hypothesize that the primary bottleneck is not the re-weighting mechanism itself,

Table 3: *Efficacy* comparison between *base* and *instruction-tuned* (it) versions of LLMs.

| LLMs | Llama3 (8B) | | Qwen2.5 (7B) | | Llama3 (8B) | | Qwen2.5 (7B) | |
|---|---|---|---|---|---|---|---|---|
| Versions | (*base*) | (*it*) | (*base*) | (*it*) | (*base*) | (*it*) | (*base*) | (*it*) |
| | ZSRE | | | | COUNTERFACT | | | |
| Original | 0.2627 | 0.2240 | 0.2016 | 0.2514 | 0.0087 | 0.0085 | 0.0078 | 0.0087 |
| IKE | 0.5233 | 0.5158 | 0.6974 | 0.8922 | 0.0055 | 0.0056 | 0.5001 | 0.7693 |
| **AdaRes** | **0.6571** | **0.7430** | **0.9264** | **0.9498** | **0.6545** | **0.5965** | **0.6475** | **0.9751** |

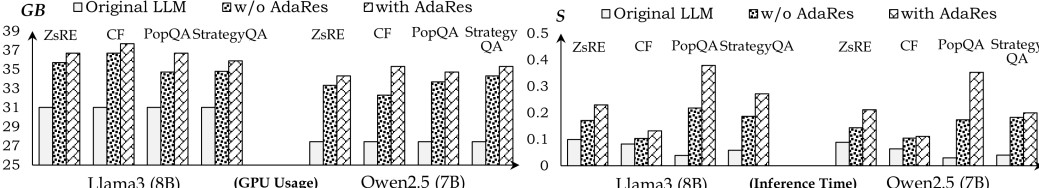

Figure 6: Efficiency comparison: GPU memory (left) and inference time (right).

but the preceding and fundamental trust estimation step. This process relies on the model's internal attention probes, and its accuracy is therefore contingent upon the model's foundational language understanding and its ability to discern contextual relevance. If the model fails to properly comprehend the input, the estimated $\alpha^{(l)}$ and $\beta^{(l)}$ will be unreliable, leading to incorrect recalibration. To test this hypothesis, we conduct a controlled experiment, *i.e.* migrating our evaluation from the *base* pre-trained models to their *instruction-tuned* versions, which are explicitly optimized to better comprehend and follow contextual instructions. The results are presented in Table 3. The findings are striking and strongly support our hypothesis. With the instruction-tuned version of Qwen2.5 (7B), the *Efficacy* score on COUNTERFACT jumps from $60^+\%$ to over $90^+\%$. This improvement, achieved simply by enhancing the base model's comprehension abilities, confirms that the conceptual framework of AdaRes is sound. The primary source of error in the *base* model lies in its inability to consistently generate an accurate trust signal in the first place, not in the *Lambda Function*'s ability to act on that signal. This is a promising result, as it implies that AdaRes will naturally become more effective as foundational models continue to improve. It also validates the key direction for future work of introducing more sophisticated trust estimators (see discussion in Limitation B).

### 4.7 EFFICIENCY ANALYSIS: GPU MEMORY USAGE AND INFERENCE LATENCY

AdaRes is a training-free mechanism that operates purely at the inference stage. While this avoids costly training, it introduces additional computations for the on-the-fly trust estimation. To provide a comprehensive assessment of its efficiency, we compare AdaRes against the standard contextual inference baseline in terms of peak GPU memory usage and end-to-end inference latency. As shown in Figure 6, AdaRes introduces only a marginal increase in both memory consumption and latency. Crucially, these measurements represent the full cost of our method, including the parallel probe streams for trust estimation, without any optimizations such as caching or pre-computation. This demonstrates that AdaRes is an inherently lightweight mechanism that can be integrated into existing inference pipelines with modest impact, confirming its practicality for real-world deployment.

### 5 CONCLUSION

In this paper, we present a preliminary exploration into mediating conflicts between contextual information and parametric knowledge in an LLM, thereby enabling more reliable utilization of contextual information. We introduce *Adaptive Residual* (AdaRes), a novel, training-free mechanism that performs dynamic, test-time trust calibration directly within the model's residual pathway. From extensive experiments, we observe several key insights: (1) the layer-wise analysis reveals a failure mode in vanilla models: their under-utilization of context often stems from either over-retrieval of

internal knowledge in lower layers (leading to the neglect of contextual signals) or insufficient attention to context during semantic fusion in middle-to-high layers (insufficient integration of contextual information during semantic fusion). (2) the most effective interventions for promoting contextual fidelity often follow a bimodal distribution, requiring simultaneous modulation of both early-layer knowledge retrieval and late-layer semantic synthesis. (3) crucially, as a pure inference-time method, AdaRes is exceptionally lightweight, achieving these gains with negligible computational overhead. This highlights its significant potential for practical applications where both reliability and efficiency are paramount.

# 6 ETHICS STATEMENT

Ours AdaRes could enhance the ability of large language models (LLMs) to adhere to contextual information, thereby improving factual grounding. However, this also introduces a potential risk: by increasing contextual fidelity, our method may make models more susceptible to generating harmful outputs if provided with misleading, malicious, or unethical content. Models using our AdaRes could thus be more prone to reproducing and propagating such material. We strongly recommend implementing robust content filtering and safety checks on all contextual inputs when using AdaRes.

# 7 REPRODUCIBILITY STATEMENT

To ensure the reproducibility of our work, we have made the complete source code and datasets available in the supplementary material. Additionally, Appendix I provides a detailed implementation guide along with a summary of key hyperparameters and further configuration details. These resources are provided to facilitate the verification of our results during the reviewing process.

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

## A    THE USE OF LARGE LANGUAGE MODELS (LLMs)

Large language models were used solely for language polishing and proofreading purposes, including refining *Abstract* and *Introduction*. All core contributions, including the idea, methodology, algorithm, experiments, and the majority of writing, were conducted entirely by the authors.

## B    LIMITATIONS

In this paper, we made a three-fold contribution: **(1) a novel conceptual perspective for mitigating knowledge conflicts by recalibrating the standard residual connection**; *(2) an attention-based mechanism for estimating usefulness of contextual versus parametric knowledge*; and *(3) a simple yet effective scaling function $\Lambda$ to apply these trust scores*. While demonstrating significant promise, we identify two main limitations that open up exciting directions for future research:

1. *Limitation in Trust Estimation*. The current method leverages the LLM's internal attention to measure the relevance of each knowledge source to the query. Although efficient and intuitive, this method is inherently self-referential, *i.e.* its reliability is contingent upon the model's own understanding capability of language. In complex scenarios where the model's foundational understanding is flawed, the attention patterns may be misleading, leading to unreliable trust estimation. A compelling direction for future work is the development of more robust and decoupled trust estimators. For instance, one could explore using specialized, external embedding models or task-specific classifiers to provide an orthogonal assessment of the query-context relevance, thereby creating a more reliable trust signal.

2. *Limitation in Trust Utilization*. This limitation concerns the expressiveness of the $\Lambda$ function. Our current implementation employs a linear interpolation, which, while simple and effective, may not be sufficient to capture complex interactions between knowledge sources This limitation points to two intriguing research directions. First, exploring more sophisticated, non-linear functions within the *Lambda* block could enable a more precise control over the re-weighting process. Second, our method currently calibrates the selected layers independently, overlooking potential cross-layer dependencies. A fascinating future direction is to design hierarchical or collaborative re-weighting schemes where trust signals are propagated or co-adjusted across layers. Such a mechanism could enable a more holistic and globally coherent reconciliation of knowledge.

These limitations highlight meaningful directions for further improving the robustness and expressivity of training-free knowledge conflict resolution.

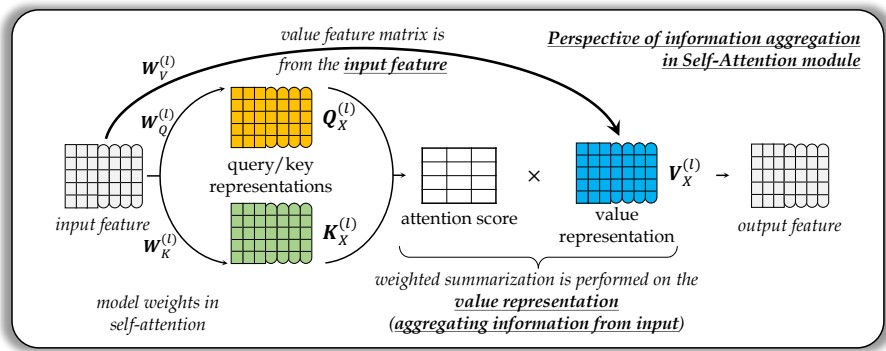

(a) Perspective of information aggregation in the `Attention` module.

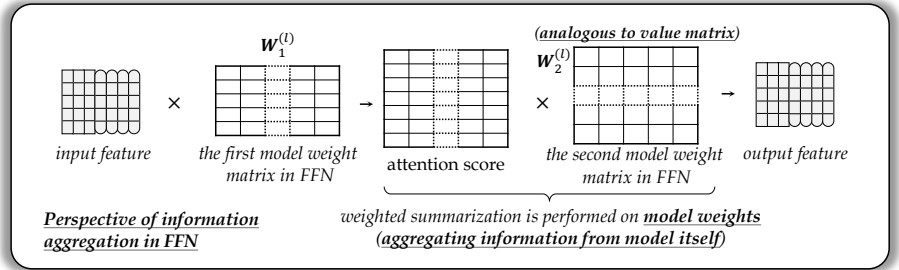

(b) Perspective of information aggregation in the `FFN` module.

Figure 7: Explanation of `Attention` and `FFN` from the perspective of information aggregation.

## C   REVISIT THE ROLE OF SELF-ATTENTION AND FEED-FORWARD NETWORK

In this paper, we build on the perspective that the `Attention` and `FFN` modules function as distinct information aggregation mechanisms: one operating on the immediate context and the other on the model's internal parametric memory. To elucidate this viewpoint, which underpins our *adaptive residual* method, we provide a detailed explanation of each module below.

### C.1   PERSPECTIVE OF INFORMATION AGGREGATION IN THE ATTENTION MODULE

This module is the primary mechanism by which LLMs dynamically route and aggregate information from the input sequence. It achieves this by computing attention scores that represent the interaction between each pair of input tokens. In a multi-head attention block, this process is parallelized to capture diverse relational patterns. Each head's operation can be formulated as:

$$\boldsymbol{S}_X^{(l)} = \text{Softmax}\left(\frac{\boldsymbol{Q}_X^{(l)}(\boldsymbol{K}_X^{(l)})^\top}{\sqrt{D_A}}\right), \quad \boldsymbol{Q}_X^{(l)} = \boldsymbol{X}^{(l)}\boldsymbol{W}_Q^{(l)}, \boldsymbol{K}_X^{(l)} = \boldsymbol{X}^{(l)}\boldsymbol{W}_K^{(l)} \tag{6}$$

Here, $\boldsymbol{Q}_X^{(l)}$ and $\boldsymbol{K}_X^{(l)}$ denote the Query and Key feature matrices which are from the input feature $\boldsymbol{X}$ projected via the attention weight matrices $\boldsymbol{W}_Q^{(l)}$ and $\boldsymbol{W}_K^{(l)}$. $\sqrt{D_A}$ is the dimension of the attention head. The resulting attention weight $\boldsymbol{S}_X^{(l)}$ quantifies the relevance of every token to every other token in the input. In our paper, we leverage these scores as a direct signal of the context's usefulness for answering a given query prompt, forming the basis of our contextual trust score.

The final output of the attention head is a weighted aggregation of Value vectors, where the weights are the attention scores $\boldsymbol{S}_X^{(l)}$:

$$\boldsymbol{X}^{(l+1)} = \boldsymbol{S}_X^{(l)}\boldsymbol{V}_X^{(l)}, \quad \boldsymbol{V}_X^{(l)} = \boldsymbol{X}^{(l)}\boldsymbol{W}_V^{(l)} \tag{7}$$

Crucially, the Value vectors $\boldsymbol{V}_X^{(l)}$ are also projections fo the input representations $\boldsymbol{X}^{(l+1)}$. Therefore, the entire self-attention operation can be viewed as a dynamic summarization or aggregation of information **drawn directly from the input context.**

## C.2 PERSPECTIVE OF INFORMATION AGGREGATION IN THE FFN MODULE

Following the `Attention` module, FFN acts as a content-based memory lookup, allowing the model to access and integrate its stored parametric knowledge. It is typically composed of two linear layers with an intermediate non-linear activation, which can also be interpreted through a *key-value* lens. The first transformation can be seen as identifying relevant knowledge patterns:

$$\boldsymbol{R}_X^{(l)} = \text{ReLU}(\underbrace{\boldsymbol{U}^{(l)}}_{query}\underbrace{\boldsymbol{W}_1^{(l)}}_{key}) \tag{8}$$

where $\boldsymbol{U}^{(l)}$ serves a Query signal. This query interacts with the first weight matrix $\boldsymbol{W}_1^{(l)}$ which can be conceptualized as the Key representing the learned pattern. The activation function determines which of these patterns are activated by the Query, producing the relevance matrix $\boldsymbol{R}_X^{(l)}$. This relevance then determines how to combine the knowledge stored in the second weight matrix:

$$\boldsymbol{F}^{(l+1)} = \boldsymbol{R}_X^{(l)} \underbrace{\boldsymbol{W}_2^{(l)}}_{value} \tag{9}$$

The weight matrix $\boldsymbol{W}_2^{(l)}$ acts as the Value store, where each row contains a piece of information. The output $\boldsymbol{F}^{(l+1)}$ is thus a weighted combination of these "Value" rows. Consequently, the FFN module can be regarded as an aggregation of the model's own **parametric memory**.

## C.3 COMPARISON BETWEEN ATTENTION AND FFN

Drawing from the *key-value* analogy to its conclusion, both modules perform a similar abstract operation: a query-activated, weighted aggregation of values. The crucial distinction lies in the sources of these values, *i.e.* generated from the input in `Attention`, pre-existing memory in a weight matrix for FFN. *This fundamental difference, depicted in Figure 7, motivates our method to separately estimate and re-weight the contributions from these two distinct knowledge sources.*

## D $\beta^{(l)}$ ESTIMATION

The trust score $\beta^{(l)}$ quantifies the relevance of the model's memory for answering the query prompt $Z$. Analogous to the estimation of $\alpha^{(l)}$ for contextual knowledge, we derive $\beta^{(l)}$ by probing the FFN at test-time without updating any parameters.

Drawing from the *key-value* perspective of FFN (as detailed in Appendix C.2), we also compute a per-token relevance score by first calculating the activation matrix $\boldsymbol{R}^{(l)}$ for the query tokens and then averaging these activations across the FFN's intermediate dimension:

$$\boldsymbol{R}^{(l)} = \text{ReLU}(\boldsymbol{U}_Z^{(l)}\boldsymbol{W}_1^{(l)}), \quad \boldsymbol{R}^{(l)} \in \mathbb{R}^{N \times D_F}; \qquad \boldsymbol{r}^{(l)} = \frac{1}{D_F}\boldsymbol{R}^{(l)}\mathbf{1}_{D_F}, \quad \boldsymbol{r}^{(l)} \in \mathbb{R}^N \tag{10}$$

Here, $\boldsymbol{U}_Z^{(l)}$ represents the input representations for the query tokens $Z$ entering the FFN block. $\boldsymbol{R}^{(l)}$ is the resulting *query*-to-*parametric memory* relevance score that reflects the potential usefulness of the parametric knowledge; $\boldsymbol{r}^{(l)}$ is the final vector of per-token relevance scores, which averages the attention mass $\boldsymbol{R}^{(l)}$ for each query token over the context span. $D_F$ is the dimensionality of the FFN's intermediate layer, which corresponds to the number of learned patterns in the parametric memory store. $\mathbf{1}_{D_F}$ is an all-ones column vector of length $D_F$. Following the same robust estimation procedure used for $\alpha^{(l)}$, the final trust value $\beta^{(l)}$ is computed by taking average on $\boldsymbol{r}^{(l)}$.

## E ALGORITHM

Algorithm 1 details the overall process of the proposed adaptive residual. Generally, our AdaRes is designed to recalibrate the trust of different knowledge sources. During the process, the core

---

**Algorithm 1** Flow of using Adaptive Residual in large language models

---

**Require:** Context $C$, Query $Z$, Input $X = [C; Z]$, $\mathcal{H}$.
**Ensure:** Model predictions $\mathcal{M}_\Theta(X)$ based on our *adaptive residual*.
 1: // Initialize representations of $C$, $Z$, $X$ from token embeddings
 2: $\boldsymbol{X}^{(0)}, \boldsymbol{C}^{(0)}, \boldsymbol{Z}^{(0)} \leftarrow$ `TokenizeAndEmbed`$(X, C, Z)$
 3: // Model Forward Pass
 4: **for** *layer_idx* in `enumerate`$(L)$ **do**
 5:    // 1. Estimate trust scores using probe representations
 6:    Estimate $\alpha^{(l)}$ via Eq. 4 based on the context feature $\boldsymbol{C}^{(l)}$ and query feature $\boldsymbol{Z}^{(l)}$
 7:    Estimate $\beta^{(l)}$ via Eq. 10 based on the query feature $\boldsymbol{Z}^{(l)}$
 8:    // 2. Apply AdaRes to the main forward pass of $X$
 9:    **if** *layer_idx* in $\mathcal{H}$ **then**
10:       // Perform AdaRes in this layer for the input feature
11:       $\boldsymbol{X}^{(l+1)} \leftarrow$ `AdaRes_layer`$(\boldsymbol{X}^{(l)}, \alpha^{(l)}, \beta^{(l)})$   ▷ Use Eq. 3
12:    **else**
13:       // Perform original transformer layer for the input feature
14:       $\boldsymbol{X}^{(l+1)} \leftarrow$ `original_layer`$(\boldsymbol{X}^{(l)})$
15:    **end if**
16:    // 3. Update probe representations for the next layer's estimation of $\alpha^{(l+1)}$ and $\beta^{(l+1)}$
17:    $\boldsymbol{C}^{(l+1)} \leftarrow$ `original_layer`$(\boldsymbol{C}^{(l)})$
18:    $\boldsymbol{Z}^{(l+1)} \leftarrow$ `original_layer`$(\boldsymbol{Z}^{(l)})$
19: **end for**
20: Logits $\leftarrow$ `LMHead`$(\boldsymbol{X}^{(L)})$
21: **return** `Decode`(Logits)

---

challenge is to obtain a clean signal of how much the *query* $Z$ relies on the *context* $C$ versus the model's *internal knowledge*, without these signals interfering with each other. To address this, AdaRes employs a *three-stream* forward pass strategy (See Lines 11-13, Line 17 and Line 18 in our algorithm). The *primary stream* processes the main input $X$ and is the sole pathway where our adaptive residual is ultimately applied to produce the final output. Crucially, running in parallel are two auxiliary "*probe streams*", which process the context and query in isolation (See Line 17 and Line 18). These probe streams are not used for the final prediction, and their sole purpose is to generate pure, unentangled representations of the context and query at every layer. As the model computes layer by layer, these clean probe representations are used to dynamically calculate the trust scores $\alpha^{(l)}$ and $\beta^{(l)}$. Armed with these instance-specific trust values, AdaRes then intervenes in the primary stream's computation for a selected set of layers (see Line 11). This design, while requiring parallel computations, ensures that our trust calibration is guided by clear and disentangled signals, facilitating a more robust and principled reconciliation of knowledge conflicts at inference time.

## F   THE $\Lambda$ FOR SYNERGISTIC KNOWLEDGE FUSION: **SCENARIO #1**

While the *asymmetric scaling function* (Appendix G) is optimized for decisive conflict resolution, a different design is needed to better handle **Scenario #1**: *Synergistic Knowledge Fusion*. **In this scenario, the contextual and parametric knowledge sources are not in direct conflict but are complementary.** The goal is not to choose one over the other, but to dynamically and smoothly blend them based on their relative relevance.

For such tasks, a **Normalized Linear Interpolation** function is more appropriate. Its design prioritizes stability and calibrated blending, particularly when the trust in both sources is balanced, formulated as:

$$\Lambda(\boldsymbol{A}^{(l)}, \boldsymbol{F}^{(l)}; \alpha^{(l)}, \beta^{(l)} | \mathcal{H}) = 2 \cdot \left( \frac{\alpha^{(l)}}{\alpha^{(l)} + \beta^{(l)}} \right) \boldsymbol{A}^{(l)} + 2 \cdot \left( \frac{\beta^{(l)}}{\alpha^{(l)} + \beta^{(l)}} \right) \boldsymbol{F}^{(l)} \qquad (11)$$

The properties of this function are distinct and ideal for fusion tasks:

- **High Confidence Behavior:** In extreme cases where one source is highly trusted over the other (*e.g.* $\alpha^{(l)} \gg \beta^{(l)}$), this function behaves similarly to our primary design in Section 3.2.2, approximating $\mathbf{2} \cdot \mathbf{A^{(1)}}$ and strongly amplifying the trusted source.

- **Balanced Trust Behavior (Key Property):** The crucial difference lies in its behavior under uncertainty or balanced trust ($\alpha^{(l)} \approx \beta^{(l)}$). In this case, the terms $\alpha^{(l)}/(\alpha^{(l)} + \beta^{(l)})$ and $\beta^{(l)}/(\alpha^{(l)} + \beta^{(l)})$ both approach 0.5. The function then becomes $2 \cdot 0.5 \cdot \boldsymbol{A}^{(l)} + 2 \cdot 0.5 \cdot \boldsymbol{F}^{(l)} = \boldsymbol{A}^{(l)} + \boldsymbol{F}^{(l)}$. This property, which we term **graceful recovery**, allows the function to seamlessly revert to the original vanilla residual block when the model is uncertain.

This design avoids imposing an aggressive bias when one is not needed, making it a more stable and suitable choice for tasks requiring nuanced integration of information rather than hard selection. **The exploration of such scenario-adaptive $\Lambda$ functions remains a promising direction for future work, as discussed in Appendix B**.

## G  DISCUSSION ON THE DESIGN OF LAMBDA FUNCTION

The primary role of the $\Lambda$ function in this work is to serve as a robust mechanism for decisive conflict resolution. In this targeted scenarios (**Scenario #2 and #4**), the model is not asked to blend knowledge, but to make a firm choice between two conflicting sources. The core challenge in this setting is the potential for noisy or unreliable trust scores ($\alpha^{(l)}, \beta^{(l)}$). A naive re-weighting function could be brittle: if the trust score for the truly correct source is erroneously estimated to be low, the function might incorrectly suppress this vital information, leading to failure.

To mitigate this risk, we designed the **Asymmetric Scaling** function (as shown in Eq.5) with a key principle in mind: *to be robust to worst-case trust estimations by providing a protective floor for the prioritized contextual knowledge source*. To further explore why we designed the function $\Lambda$ like Eq.5, let us analyze its behavior in extreme circumstances:

- **Best Case (High Contextual Trust):** When the context is correctly identified as highly trustworthy ($\alpha^{(l)} \gg \beta^{(l)}$), the term $\alpha^{(l)}/(\alpha^{(l)} + \beta^{(l)})$ approaches 1. The function then approximates $2 \cdot \boldsymbol{A}^{(l)} + 0 \cdot \boldsymbol{F}^{(l)}$, strongly amplifying the contextual stream while completely suppressing the conflicting parametric stream. This achieves the ideal outcome.

- **Worst Case (Low Contextual Trust):** If the trust estimation fails and yields a very low score for the context ($\alpha^{(l)} \ll \beta^{(l)}$), the term $\alpha^{(l)}/(\alpha^{(l)} + \beta^{(l)})$ approaches 0. The function then approximates $1 \cdot \boldsymbol{A}^{(l)} + 1 \cdot \boldsymbol{F}^{(l)}$. This is the crucial protective mechanism: instead of penalizing or suppressing the contextual information, the function gracefully degrades to the **original vanilla residual block**. This ensures that the prioritized knowledge source is never actively suppressed, even with unreliable trust scores.

This design guarantees that the coefficient for the prioritized source is bounded within $[1, 2]$, while the suppressed source's coefficient is bounded within $[0, 1]$. This intentional asymmetry provides a robust bias towards the desired outcome in a conflict scenario, making it highly effective for the hard-selection tasks investigated in this paper.

Moreover, another aspect worth discussing is *how to steer the model to trust its internal parametric knowledge*. Homoplastically, the design in Eq. 5 can be adapted into the following form:

$$\Lambda(\boldsymbol{A}^{(l)}, \boldsymbol{F}^{(l)}; \alpha^{(l)}, \beta^{(l)} | \mathcal{H}) = (1 - \frac{\beta^{(l)}}{\alpha^{(l)} + \beta^{(l)}}) \, \boldsymbol{A}^{(l)} + (1 + \frac{\beta^{(l)}}{\alpha^{(l)} + \beta^{(l)}}) \, \boldsymbol{F}^{(l)}, \quad l \in \mathcal{H} \quad (12)$$

This scaling scheme now functions to protect the internal knowledge, which is not only fully consistent with the previous discussion on safeguarding contextual knowledge but also forms a symmetric counterpart to Eq. 5. This further demonstrates the highly generic and compatible nature of our method. **We term this scaling mechanism the *Lambda Function*, intending to highlight its convenience and ease of use, much like `lambda expressions` in programming languages such as `Python`, which allow for concise and flexible function definitions.**

## H    Detailed Discussion of Related Work

### H.1    Analysis: LLMs' Behavior Under Conflicts

Contextual knowledge injection typically materializes as retrieval-augmented generation, which conditions generation on retrieved information (Wu et al., 2024b; He et al., 2024; Wang et al., 2025b). It is important to know *when discrepancies between contextual and parametric knowledge arise, how does the model behave?* since unpredictable results degrade reliability. Early investigations in extractive and open-domain QA reported divergent outcomes. Longpre et al. (2021) examine models under QA scenarios and find a tendency to over-rely on parametric knowledge. Revisiting a similar setup, Chen et al. (2022) observe the opposite pattern in their best-performing configurations, namely a predominant reliance on contextual evidence. For LLMs, more recent studies further nuance this picture. Tan et al. (2024) examine how LLMs blend retrieved context with generated knowledge in the open-domain QA setup and report that models often favor parametric knowledge when retrieval yields incomplete evidence, particularly under conflicting sources. In contrast, by generating controlled conflicts between memorized facts and curated external context, Xie et al. (2024) show that models can be highly receptive to contextual information, even against their parametric beliefs. Complementarily, Qian et al. (2024) evaluate interactions between parametric knowledge and external knowledge graphs and find that models frequently deviate from parametric recall when confronted with direct conflicts or fine-grained contextual changes. Farahani & Johansson (2024) indicate that in cases where the model can choose between both types of information (parametric and nonparametric), it relies more on the context than the parametric knowledge. Under interactive, multi-turn settings, Xu et al. (2024a) find a preference of LLMs for logically structured presentations, even when such structure conflicts with factual accuracy. Taken together, what is the conclusion? The literature indicates *no universal rule for whether an LLM prioritizes contextual or parametric knowledge* (Xu et al., 2024b). These findings motivate mechanisms that adapt reliance on external versus parametric sources at inference time, rather than enforcing a fixed policy.

### H.2    Solutions: Handling Conflicts

To reconcile contextual and parametric knowledge, prior works can be organized into four categories Xu et al. (2024b): *context-first alignment*, *parametric-preservation strategies*, *conflict detection with a dual-response strategy*, and *evidence fusion*. **Context-first** alignment align generation with the provided context and explicitly prioritize external evidence. Gekhman et al. (2023) maintain factual consistency with source documents by annotating model-generated summaries using LLM-based teachers. Xue et al. (2023) improve alignment with factual knowledge through direct knowledge enhancement and reinforcement learning. Zhou et al. (2023) design specialized prompting strategies that strengthen adherence to contextual evidence and yield gains on context-sensitive tasks. Shi et al. (2024) introduce Context-aware Decoding to amplify the distributional difference between decoding with and without context, thereby encouraging reliance on the injected evidence. **Parametric-preservation** strategies preserves the model's internal memory when conflicts arise, either by modifying parameters or by reducing misinformation in the context. Knowledge editing methods directly rewrite parameters to replace or refine stored facts, eliminating specific discrepancies (Meng et al., 2022; 2023; Wang et al., 2024a; Xu et al., 2025; Zhai et al., 2025a; Fang et al., 2025). Although effective, such changes are persistent and may introduce collateral effects. To keep predictions faithful to internal knowledge without editing, several works reduce contextual noise or verify parametric beliefs before answering. Examples include misinformation detection and vigilant prompting defenses (Pan et al., 2023), and a system prompt that instructs the model to verify memorized knowledge prior to response generation in interactive settings (Xu et al., 2024a). Credibility-aware attention modification further adjusts influential attention heads using document credibility estimates to downweight low-quality retrieved content (Deng et al., 2025). **Dual-response** strategy first detect the presence and type of conflict, then invoke tailored resolution strategies. Wang et al. (2024b) develop a three-step process that identifies conflicts and produces distinct, informed responses according to the detected discrepancy. **Fusion** methods aim to merge information from both sources. Zhang et al. (2023) train discriminators on silver labels to pair compatible generated and retrieved passages for joint use. Jin et al. (2024a) propose a contrastive decoding algorithm that maximizes differences between logits under conflicting inputs and calibrates confidence toward the truthful answer.

```
adares/
├── adares_llms          (directory of LLMs that applies our AdaRes)
│   ├── adares_modelling_gemma3.py
│   ├── adares_modelling_llama.py
│   ├── adares_modelling_phi3.py
│   └── adares_modelling_qwen2.py
├── adares_main.py  (main file of AdaRes)
├── data
├── scripts              (Running Scripts)
└── utils.py             (miscellaneous utility function in AdaRes)
```

Figure 8: Code directory of our implementation.

## I    IMPLEMENTATION DETAILS

This section provides a comprehensive overview of the proposed method and specific configurations used in our experiments to ensure full reproducibility. Our source code, built upon PyTorch and the Hugging Face Transformers library, is attached to the supplementary material. A map of the code directory structure is also provided in Figure 8.

### I.1    MODELS AND ENVIRONMENT

Our experiments leverage a diverse suite of publicly available large language models to demonstrate the broad applicability of AdaRes. All models were used with their original pre-trained weights without any further fine-tuning. The specific models are: **meta-llama/Meta-Llama-3-8B**, **Qwen/Qwen2.5-7B**, **meta-llama/Llama-3.3-70B-Instruct**, **Qwen/Qwen3-32B**, **microsoft/Phi-3-mini-4k-instruct** (3.8B), **microsoft/Phi-3-medium-4k-instruct** (14B), **google/gemma-3-4b-it**, **google/gemma-3-12b-it**. Our experimental environment was configured with `Python 3.9`, `PyTorch 2.4`, and Hugging Face `Transformers 4.55`. All experiments were conducted on NVIDIA A100 GPUs with 80GB of VRAM.

### I.2    SUMMARY OF ADARES CONFIGURATIONS OF THIS PAPER

The core hyperparameters for AdaRes are the set of layers $\mathcal{H}$ (*i.e. where it is applied*) and the number of query tokens used for trust estimation *Top-n*. We summarize the specific configurations for each model in Table to ensure our results are easy to reproduce. The only one configuration need to clarify is **Layer Selection** ($\mathcal{H}$) which determines which layers are selected to apply the AdaRes. We tested different settings on different LLMs to seek the optimal performance, which are listed as follows:

- **Llama3 (8B)**: $\mathcal{H} = \{6, 16, 17, 25\}$ for ZsRE, $\mathcal{H} = \{3, 12, 17, 25\}$ for COUNTERFACT, $\mathcal{H} = \{6, 10\}$ for ConflictQA-PopQA, $\mathcal{H} = \{9, 17\}$ for ConflictQA-StrategyQA.

- **Llama3 (8B) Instruct**: $\mathcal{H} = \{16, 30, 31\}$ for ZsRE, $\mathcal{H} = \{5, 9, 17, 31\}$ for COUNTERFACT.

- **Qwen2.5 (7B)**: $\mathcal{H} = \{1, 4, 22\}$ for ZsRE, $\mathcal{H} = \{3, 24, 26, 27\}$ for COUNTERFACT, $\mathcal{H} = \{5, 12\}$ for ConflictQA-PopQA, $\mathcal{H} = \{6, 16\}$ for ConflictQA-StrategyQA.

- **Qwen2.5 (7B) Instruct**: $\mathcal{H} = \{1\}$ for ZsRE, $\mathcal{H} = \{1\}$ for COUNTERFACT.

- **Phi3 (3.8B)**: $\mathcal{H} = \{10, 15, 20, 26, 30\}$ for ZsRE, $\mathcal{H} = \{17, 20, 21, 26, 27, 30\}$ for COUNTER-FACT, $\mathcal{H} = \{20\}$ for ConflictQA-PopQA, $\mathcal{H} = \{17\}$ for ConflictQA-StrategyQA.

- **Phi3 (14B)**: $\mathcal{H} = \{0, 8, 16, 20, 28, 30, 36\}$ for ZsRE, $\mathcal{H} = \{16, 24, 25, 36, 37\}$ for COUNTER-FACT, $\mathcal{H} = \{22\}$ for ConflictQA-PopQA, $\mathcal{H} = \{14\}$ for ConflictQA-StrategyQA.

- **Gemma3 (4B)**: $\mathcal{H} = \{17\}$ for ZsRE, $\mathcal{H} = \{17\}$ for COUNTERFACT, $\mathcal{H} = \{7\}$ for ConflictQA-PopQA, $\mathcal{H} = \{16\}$ for ConflictQA-StrategyQA.

- **Gemma3 (12B)**: $\mathcal{H} = \{27\}$ for ZsRE, $\mathcal{H} = \{9\}$ for COUNTERFACT, $\mathcal{H} = \{19\}$ for ConflictQA-PopQA, $\mathcal{H} = \{17\}$ for ConflictQA-StrategyQA.

### I.3 REPRODUCE EXPERIMENTAL RESULTS

To facilitate direct replication of our main results, we provide execution scripts in the `scripts/` directory of our supplementary code. Prior to execution, users should download the official model weights from Hugging Face and update the corresponding paths in the provided configuration files.

## J ADDITIONAL EXPERIMENTS

### J.1 EXPERIMENTAL SETTINGS OF CONTEXT-FIRST EXPERIMENTS

### J.2 EVALUATION METRICS ON ZSRE AND COUNTERFACT

After finishing all editing requests, each method is assessed on $\mathcal{E}$. Following previous research, we also adopt three fundamental editing metrics in this task for evaluation.

**Efficacy** computes the success rate of editing operations:

$$Efficacy(\mathcal{E}|\Theta^*) = \sum_{i=1}^{|\mathcal{E}|} \mathbb{1}(f_{\Theta^*}(x_i) == y_i^*) \, / \, |\mathcal{E}| \tag{13}$$

where $\mathbb{1}(\cdot)$ measures the ratio at which the model predictions matched the desired outputs for each editing.

**Generality** measures whether the edited model could proceed with the semantically equivalent rephrase of each edit:

$$Generality(\mathcal{E}|\Theta^*) = \sum_{i=1}^{|\mathcal{E}|} \mathbb{1}(f_{\Theta^*}(x_i^{'}) == y_i^*) \, / \, |\mathcal{E}| \tag{14}$$

**Locality** refers to the degree to which other irrelevant knowledge has changed after editing:

$$Locality(\mathcal{K}|\Theta^*) = \sum_{j=1}^{|\mathcal{K}|} \mathbb{1}(f_{\Theta^*}(z_j) == y_j) \, / \, |\mathcal{K}| \tag{15}$$

where $\mathcal{K}$ is the knowledge set unrelated with $\mathcal{E}$, and $\mathcal{K} \cup \mathcal{E} = \emptyset$. $z_j$ is the textual prompt of one specific knowledge $k_i$ in $\mathcal{K}$, and $y_j$ is the original model prediction.

### J.3 EVALUATION METRICS ON CONFLICTQA

For ConflictQA datasets, only the *Efficacy* is used, which has the same definition with Eq. 13.

### J.4 BASELINE ON ZSRE AND COUNTERFACT

We use the *EasyEdit*[3] toolkit for all baseline model editing operations, utilizing hyperparameters recommended in its official documentation. The specific settings are detailed as follows:

#### J.4.1 FT-C

Knowledge editing of FT-C is executed at layer 21 for GPT-J, and Llama3, and 27 for Qwen 2.5 where optimization proceeds for 25 steps with a learning rate of $5e^{-4}$. The batch size is set to 1 and weight_decay is set to 0.

#### J.4.2 ROME

For ROME, Knowledge editing is performed at layer 5 for GPT-J, Llama3 and Qwen 2.5. The learning rate is $5e^{-1}$, with the optimization proceeding 25 steps for Llama3 and Qwen 2.5, and 20 steps for GPT-J. The weight-decay is set to $1e^{-3}$ for Llama3 and Qwen 2.5, and 0.5 for GPT-J. The KL factor is fixed at 0.0625 for three LLMs. Covariance statistics are collected in float32 on Wikitext using a sample size of $100,000$.

---

[3]Official Website: https://github.com/zjunlp/EasyEdit

### J.4.3 MEMIT

Knowledge updating of MEMIT is executed at layer $[4, 5, 6, 7, 8]$ for Llama3 and Qwen 2.5, and $[3, 4, 5, 6, 7, 8]$ for GPT-J. Optimization proceeds for 25 steps with a learning rate of $5e^{-1}$ for three models. The weight-decay is set to $1e^{-3}$ for Llama3 and Qwen 2.5, and $0.5$ for GPT-J. The KL factor is set to $0.0625$ for three models. Covariance statistics are also collected in float32 precision on Wikitext using a sample size of $100,000$.

### J.4.4 KN

For KN, the lr_scale = 1.0, n_toks = 10, batch_size = 1, num_steps = 20, adaptive_threshold = 0.1.

### J.4.5 PMET

For PMET, Knowledge editing is executed across layers $[3, 4, 5, 6, 7, 8]$ for GPT-J and $[4, 5, 6, 7, 8]$ for Llama3 and Llama2. The optimization process involves 30 steps for GPT-J and 20 steps for Llama3. The learning rate is set to $2e^{-1}$ for GPT-J and $5e^{-1}$ for Llama3. The weight decay of three models is configured to $0.5$. The KL factor is set to 1 for GPT-J and $0.0625$ for Llama3. Covariance statistics are also collected in float32 on Wikitext using a sample size of $100,000$.

### J.4.6 ALPHAEDIT

Knowledge updating of AlphaEdit is executed at layer $[4, 5, 6, 7, 8]$ for Llama3 and Qwen 2.5, and $[3, 4, 5, 6, 7, 8]$ for GPT-J. Optimization proceeds for 25 steps with a learning rate of $1e^{-1}$ for GPT-J and Llama3, $5e^{-1}$ for Qwen 2.5. The weight-decay is set to $0.5$ for Llama3 and GPT-J, $1e^{-3}$ for Qwen 2.5. The KL factor is set to $0.0625$ for three models. Covariance statistics are also collected in float32 precision on Wikitext using a sample size of $100,000$.

### J.4.7 LoRA

Knowledge editing of LoRA refers to AdaLoRA for all models, where optimization proceeds for 70 steps with a learning rate of $5e^{-4}$. The weight_decay is set to 0, KL_factor = 0, rank = 8, LoRA_Alpha = 32, and LoRA_dropout = 0.1.

### J.4.8 R-ROME

For R-ROME, Knowledge editing is performed at layer 5 for GPT-J, Llama3 and Qwen 2.5 models. The learning rate is $5e^{-1}$, with the optimization proceeding 25 steps for Llama3 and Qwen 2.5, and 20 steps for GPT-J. The weight-decay is set to $1e^{-3}$ for Llama3 and Qwen 2.5, and $0.5$ for GPT-J. The KL factor is fixed at $0.0625$ for both models. Covariance statistics are collected in float32 on Wikitext using a sample size of $100,000$.

### J.4.9 WISE

Knowledge editing is performed at layer 23 for Qwen 2.5 (7B). Hyperparameters are configured as follows: *mask_ratio* is set to 0.2 with *edit_lr* = 1.0 and *norm_constraint* = 1.0. The *act_margin* is set to $[5.0, 20.0, 10.0]$ and *act_ratio* = 0.88. The *merge_freq* is set to 1000 with *merge_alg* being *'ties'*. The *densities* is 0.53 and *weights* is 1.0.

### J.4.10 GRACE

Knowledge editing is performed at layer 18 for Qwen 2.5 (7B). Additionally, *edit_lr* and *n_iter* are 1.0 and 50, respectively; with *eps* being 1.0. The *dist_fn* is set to *euc*. *val_init* is *cold* witt *val_train* being *sgd*. *reg* is *early_stop*. The *eps_expand* is set to *coverage* and *num_pert* is 8.

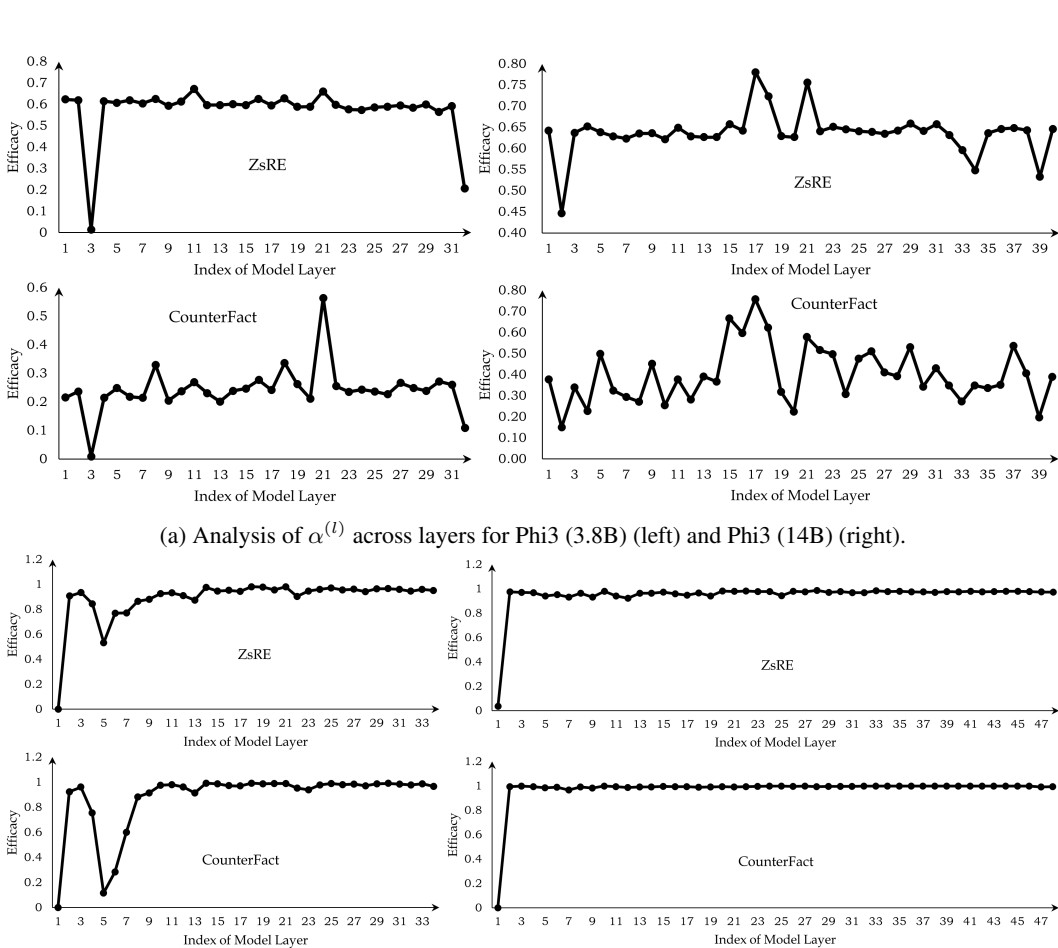

(a) Analysis of $\alpha^{(l)}$ across layers for Phi3 (3.8B) (left) and Phi3 (14B) (right).

(b) Analysis of $\alpha^{(l)}$ across layers for Gemma3 (4B) (left) and Gemma3 (12B) (right).

Figure 9: Trend of $\alpha^{(l)}$ across layers for different LLMs.