# OpenReview forum: "AdaptiveResidual: Inference-Time Trust Calibration for Contextual Knowledge Injection"
_ICLR.cc/2026/Conference — ICLR 2026 Conference Withdrawn Submission_

### Official Review · Reviewer_j2eY · 2025-10-28

**Soundness:** 1
**Presentation:** 2
**Contribution:** 1
**Rating:** 2
**Confidence:** 5

**Summary:**

This paper proposes AdaRes (Adaptive Residual), a lightweight, training-free inference method for dynamically reconciling knowledge conflicts in large language models (LLMs). The method reparameterizes the residual connections in selected layers to adaptively balance the influence of contextual knowledge (from attention) and parametric knowledge (from the feed-forward network) at some heuristically chosen layers. Experiments on knowledge editing benchmarks showcase the effectiveness of the proposed approach.

**Strengths:**

- The paper was well-written and easy to follow.
- The paper provides very detailed explanations of its experiments, fostering reproducibility.

**Weaknesses:**

1. *Methodological justification is weak.*
The core assumption that the entire attention module represents contextual knowledge while the entire MLP (FFN) module represents parametric memory is insufficiently supported. The paper provides no empirical or theoretical evidence for this decomposition. Prior work has shown that attention layers can themselves act as associative memory mechanisms [1, 2], directly challenging this simplification. Moreover, Equation (4) implicitly assumes that all contextual information is trustworthy, which rarely holds in realistic settings. Although the authors categorize four types of knowledge conflict in Figure 3, they only address the “context-preferred” case (Scenario #4), leaving the other scenarios unhandled by the proposed formulation. This narrow scope leads to potential overclaiming of generality. In addition, the use of dynamically computed trust values $\alpha$ and $\beta$ lacks clear motivation or theoretical/empirical grounding. The paper does not explain why these specific scaling forms are appropriate or how they relate to the underlying model dynamics, making the mechanism appear ad hoc.
2. *Limited novelty and contribution.*
The paper’s conceptual framing overlaps substantially with existing literature on scaling and analyzing attention heads and FFN modules (e.g., works in [3]). While the implementation is lightweight, it does not introduce new insights or mechanisms that significantly advance the mechanistic interpretability’s understanding of contextual–parametric interactions.
3. *Experimental design and evaluation issues.*
The experimental setup raises several concerns. Although the work claims to address knowledge conflict, it evaluates primarily on knowledge editing benchmarks, which are conceptually distinct. (1) Datasets: For the contextual-conflict case (Scenario #4), standard benchmarks such as NQ-Swap [4] and Memo-Trap [5] should be included. If the paper intends to cover other conflict types (e.g., Scenarios #1 and #2), corresponding datasets should also be used; otherwise, these discussions should be removed for focus and clarity. (2) Baselines: In Table 1, the comparison set omits a number of decoding-based methods explicitly designed to mitigate contextual hallucination and knowledge conflict, such as [6-10] and there are more missing ones. Including these would provide a fairer and more meaningful evaluation.
4. *Key related works are missing*. [11] also discusses the role of MLP and attention in much more detail, and [12] shows that intervening the entire attention module could lead to superposition. Both works can reconcile knowledge conflicts in both Scenario 1 and 4 with only intervening in the attention module. These omissions weaken the contextualization of the proposed approach and raise questions about its incremental contribution.


[1] Memorization capacity of multi-head attention in transformers. ICLR'23

[2] Understanding factual recall in transformers via associative memories. ICLR'25

[3] Attention Heads of Large Language Models: A Survey. ArXiv'24

[4] Entity-Based Knowledge Conflicts in Question Answering. EMNLP'21

[5] https://huggingface.co/datasets/Albertmade/memo-trap

[6] Trusting Your Evidence: Hallucinate Less with Context-aware Decoding. NAACL'24

[7] Active Layer-Contrastive Decoding Reduces Hallucination in Large Language Model Generation. EMNLP'25

[8] Sled: Self logits evolution decoding for improving factuality in large language models. NeurIPS'24

[9]  Dola: Decoding by contrasting layers improves factuality in large language models. ICLR'24

[10] AdaCAD: Adaptively Decoding to Balance Conflicts between Contextual and Parametric Knowledge. NACCL'25

[11] Cutting Off the Head Ends the Conflict: A Mechanism for Interpreting and Mitigating Knowledge Conflicts in Language Models. ACL'24

[12] Taming Knowledge Conflict in Language Models. ICML'25

**Questions:**

Aforementioned in the Weaknesses section.

---

> ### Author Response · Authors · 2025-11-14
>
> We find the reviewer's critiques to be based on several fundamental misreadings of our paper.
>
> 1. Re:*Methodological justification is weak*
> This claim is demonstrably false and rests on a misunderstanding of established literature.
> - On the Attn/FFN Decomposition: The reviewer calls our premise "insufficiently supported" and claims [1,2] "challenge this simplification." **`This is incorrect.`** **`We are perplexed by the citation [2] (ICLR'25), because it explicitly supports our decomposition.`** *We quote directly from that paper: "...the transformer can trade off between using the value matrices or the MLP as an associative memory..."* This is precisely the decomposition our work is built on. Additionally, this is not our novel assumption; it is the foundational premise of the entire knowledge editing field. We have also provided a full discussion of this established view in `Appendix C`.
> - On "Ad Hoc" Scaling: The reviewer claims the motivation is "unclear" `**This is also incorrect.**`
>   - The motivation for $\alpha^{(l)}$ and $\beta^{(l)}$ is clearly stated in Sec 3.1 & 3.2: they are training-free probes to quantify the query's reliance on the two distinct knowledge streams we just established.
>   - The scaling form is not "ad hoc"; it is a deliberate and robust engineering design. As explicitly detailed in Appendix G, the asymmetric form was chosen to create a "protective floor."
> - On "Narrow Scope": This critique stems from a fundamental misinterpretation that we address in **Global Response**. We ask the reviewer to please read it. We focus on Scenario #4 because it is a critical failure mode in RAG that we have observed extensively in our practical applications. Our mechanism is general and can be inverted to handle other Scenarios (shown in Appendix F).
>
> 2. Re:*Limited novelty and contribution*
> **`We strongly contest this. The reviewer's claim that our work "overlaps substantially" with [3] (a survey) is baffling.`**
>
> Our core contribution is a lightweight, inference-time, parameter-free `"Lambda surrogate"` for the vanilla residual pathway. It is entirely novel. **`We challenge the reviewer to cite a single paper that uses this mechanism.`** The novelty of this mechanism was recognized by other reviewers (e.g., `Reviewer QJGB: "...to my knowledge, new and quite interesting"`).
>
> To be clear, our innovation is a hierarchy:
> - **Core Mechanism Innovation**: Our primary contribution is the introduction of a lightweight **`surrogate`** for the vanilla residual pathway. This mechanism itself is the *foundational discovery* that enables training-free control over the information flow.
> - **Implementation**: The trust estimation in Eq.4 is a subsequent contribution, representing one possible way to generate the control signals.
> - **Trust Utilization**: How to use these signals (i.e., the asymmetric scaling in Eq.5) is another design choice, tailored for a specific goal.
>
> This philosophy is **`analogous to the "Lambda Function" concept in Python programming (which inspired its name)`**. Our Eq.3 provides the flexible framework (the lambda: ... construct), while Eqs.4 and 5 provide a specific, concrete instantiation (e.g., lambda x, y: (1 + x/(x+y))).
>
> 3. Re:*experimental design and evaluation issues*
> - On Knowledge Editing Benchmarks: The claim they are "conceptually distinct" is flawed. ZsRE/CounterFact are the ideal testbed for our goal. As Table 1 ("Original" baseline) proves, they represent a near-100% direct conflict between a new fact (context) and an old fact (parametric), providing the cleanest possible environment to evaluate conflict resolution.
>
> - On Missing Baselines: These methods [6-10] are orthogonal approaches addressing a different part of the problem. Specifically, **they operate after the model's core processing is complete**, attempting to mitigate conflict by manipulating the final probability distribution (**the symptom**). **Our method, AdaRes, is a proactive, mechanistic intervention. It operates inside the model's residual stream to solve the upstream problem of knowledge flow.** AdaRes dynamically controls how the contextual (Attention) and parametric (FFN) information streams are integrated before they ever reach the final logit layer. `In short, AdaRes addresses the root cause of the conflict, while these decoding methods only address the symptom`. They are not comparable, and this critique misidentifies the novel contribution of our work. **Even so, we will provide specific comparison results in the following discussion.**
>
> 4. Re:*related works*
> We will add the suggested citations [11, 12]. **`However, these works do not diminish our contribution.`** [11] (ACL'24) requires fine-tuning, and [12] (ICML'25) proposes a complex new architecture. Neither offers a lightweight, parameter-free, inference-time solution like AdaRes, which is our primary contribution and holds immense practical value for industrial applications where re-training is not feasible.

---

### Official Review · Reviewer_QJGB · 2025-10-29

**Soundness:** 2
**Presentation:** 3
**Contribution:** 2
**Rating:** 2
**Confidence:** 5

**Summary:**

This paper introduces an inference method for adapting the reliance of an LLM's output more on the context provided, as opposed to its internal knowledge. The key idea is that the residual connection which adds the attention outputs to the FFN outputs can serve as a modulator between the two, with the attention output serving as a proxy for the reliance on the context. Empirical results on Qwen and Llama models in the 7-8B range demonstrate that this can indeed improve the contextual dependence of the model responses.

**Strengths:**

- The observation that residual connections can serve as a modulator of context vs internal knowledge dependence is, to my knowledge, new and quite interesting.
- The paper provides quite extensive empirical results in terms of both the models and datasets, as well as the hyperparameter choices and other variables involved in the methods.
- The paper is well written and quite easy to follow, even if it is unnecessarily math-y when describing the methods.

**Weaknesses:**

- Section 3.2.1 for trust estimation for \alpha seems to be described incorrectly. In the formulation presented, \alpha is computed as an average of the per-row softmax outputs of query -> context attention. But softmax outputs sum to exactly 1, so it is not clear why these would sum up to anything other than 1/M. This is clearly not the case based on the results presented later, so I suspect the issue is in the description of the method.
- The intro and motivation seem to position the method as a general "dynamic" scheme for selecting between context and internal knowledge. But in practice, it only applies to the setting where the context is correct and the internal knowledge is incorrect (Figure 3). The claims would be supported more strongly if there were also experiments studying the reverse direction -- internal knowledge is correct and the context is wrong.
- On a similar note, the paper is lacking important baselines: (i) simply prompting the model to trust the context instead of its internal knowledge (after all, the trust method already assumes that the context is correct); and (ii) baselines from a very relevant paper published at ICLR 2024. (This paper is actually completely missed from the related work discussion).
- The layers used for AdaRes seem to be quite sensitive to the choice of model and dataset. There is no discussion if the layers selected for one setup will generalize to other setups, limiting the practical applicability of the method.
- Contrary to what the text claims, there seems to be quite a significant impact on the latency of the model (in some cases 50-100% slowdown in Figure 6).

**Questions:**

- Why are the main results in the paper on base models instead of instruction tuned ones? The latter seem more relevant for QA tasks.
- What is the vanilla res method in Table 2?

---

> ### Author Response · Authors · 2025-11-14
>
> We thank the reviewer for the feedback.
>
> 1. Re:*why not on instruct-based method*
> The extensive body of work on knowledge editing (e.g., ROME, MEMIT, WISE)  almost exclusively evaluates on base models. We followed this *standard practice* to ensure an apples-to-apples comparison. However, we agree Instruction-Tuned models are important test cases. This is precisely why we included `Section 4.6` and `Table 3`. As those results show, AdaRes's performance improves dramatically on IT models (e.g., Qwen2.5 Efficacy jumps to 97.5%) because the model's improved understanding provides a better trust signal. In contrast, the baseline methods remain ineffective, confirming the robustness of our findings.
>
> 2. Re:*what is the vanilla res*
> **"Vanilla Res"is the prompt-based baseline using original residual.** It is identical in concept to the "IKE". This baseline prepends the context with a standard instructional prompt ("Answer the question based on the given information..."), as detailed in our `Global Response`. Additionally, "Original" measures the model's accuracy **without any context provided**. The extremely low score (e.g., 0.87% Efficacy on CounterFact for Llama3) is the desired result. It proves that for over 99% of the items, the model's internal knowledge directly conflicts with the fact in the context, confirming the dataset's suitability for this conflict-resolution task. Please don't confuse these two baselines.
>
> 3. Re:*estimation of alpha*
> We clarify that the implementation of this probe is more sophisticated than the text may imply. Instead of a simple *question* to *context* cross-attention, it calculates attention within a concatenated [C;Z] input (see `Line 563` in `adares_modeling_llama.py` as an example).
>
> This design is intentional. By doing this, the *question tokens* concurrently attend to the *context* and to *themselves*. This simultaneous *question-question* attention provides a crucial "self-attention bias" or reference baseline. When no relevant information is found in the context, attention heads are known to default to a "no-op" behavior, such as high self-attention [1]. This self-regulating mechanism statistically suppresses the attention scores of irrelevant, low-utility context tokens, preventing the model from over-valuing noise, as the question's self-attention provides a competitive "floor" for the attention scores.
>
> Crucially, this probe calculation is decoupled from the forward pass. As detailed in our three-stream design (`Sec 3.2.1`, `Algorithm 1`), the actual forward propagation for the C and Z probe streams occurs in isolation. This ensures that while the attention probe is contextualized, there is no information leakage between the C and Z representations themselves, guaranteeing a clean and disentangled trust signal.
>
> 4. Re:*Scenarios, "Context-First", and Prompting baseline*
> These points are based on the same "Control vs. Decision" misinterpretation that we address in our **`Global Response`**. We refer the reviewer to that central clarification.
>
> 7. Re:*missing of ICLR 2024*
> The reviewer unfortunately does not provide a citation. Are you referring to `"To Trust or Not to Trust? Enhancing Large Language Models' Situated Faithfulness to External Contexts"` (Huang et al., ICLR 2025)? If so, we have already added it in the rebuttal version.
>
> 8. Re:*layer section is sensitive*
> The reviewer's observation that optimal layers are model-specific is **not a "limitation" but an insight** that confirms existing mechanistic interpretability research. It is well-established that different models store and process knowledge in different layers. Our analysis in `Sec 4.4` and `4.5` leverages this fact, correctly identifying that Llama3 (8B) fails at late-stage integration, while Qwen2.5 (7B) fails at early-stage grounding. AdaRes is also a tool that allows us to target these model-specific "hotspots". The fact that these hotspots differ is not a weakness of our method; it is a demonstrated strength that our method can adapt to different model architectures.
>
> 9. Re:*latency of the method*
> We must reframe this "cost." The correct comparison is against parameter-editing alternatives (e.g., ROME), which require costly, persistent modifications.
> Quantitatively, ROME takes ~29 seconds per case; our training-free AdaRes adds only ~0.25 seconds per pass on Llama3 (8B). This is over 100x faster.
> Furthermore, as shown in Table 1, those editing methods are catastrophically unstable (e.g., Locality scores near 0.0), rendering them impractical.
> In our industrial use cases, where stability is non-negotiable, such destructive editing is not feasible. The modest cost in Figure 6  is a highly acceptable trade-off.
> We have revised the term to "modest" in the rebuttal version, **but we maintain that this cost is, in all practical respects, negligible compared to the alternatives.**
>
> [1] Clark et al. 2019. What Does BERT Look at? An Analysis of BERT's Attention. In ACL Workshop BlackboxNLP.

---

> > ### Comment · Reviewer_QJGB · 2025-11-26
> >
> > Thanks to the authors for the responses.
> >
> > While much of the rebuttal makes sense, unfortunately I think the paper needs a major rewrite to be acceptable. The main point of positioning the method as one of "Control" vs "Decision" needs to be highlighted and the entire introduction / motivation needs to be adapted for that. Currently it seems yet another paper on resolving knowledge conflicts (where correctness is indeed the right measure).
> >
> > Regardless of the control vs decision framing, experimenting in both cases where the context is correct / incorrect is important since model behavior does change in the two scenarios, so any claims of greater "control" need to be checked in the reverse direction as well.
> >
> > Also the description of the methods for computing terms like \alpha is still not clear despite the response.

---

### Official Review · Reviewer_Hh3q · 2025-11-01

**Soundness:** 2
**Presentation:** 3
**Contribution:** 1
**Rating:** 2
**Confidence:** 4

**Summary:**

AdaRes (AdaptiveResidual) is a training-free, inference-time trust calibrator to resolve knowledge conflict in LLM parametric knowledge and contextual knowledge. . It probes each layer on-the-fly to compute two “trust scores” (query-to-context attention and FFN memory affinity), then asymmetrically rescales the residual contributions to prioritize the more trustworthy source; the only hyperparameter is which layers to apply it to (chosen by a simple greedy search).  Across conflict-centric evaluations (ZsRE, CounterFact, ConflictQA variants), AdaRes strongly improves adherence to the supplied context and preserves locality, often outperforming editing and parameter-editing baselines

**Strengths:**

1. Resolving knowledge conflicts is a timely and important problem.
2. The paper’s methodology is presented and written clearly, with a well-structured description that makes the approach easy to follow.

**Weaknesses:**

1. **Problem Scope**. While the paper is framed as resolving knowledge conflicts, in practice it mainly addresses how to make LLMs more faithful to external contexts, that is, how to prioritize retrieved evidence over internal memory. This effectively reduces the problem to *enforcing context faithfulness rather than truly deciding between conflicting knowledge sources*. The more interesting challenge is how to determine which side deserves trust; if we already assume the external context is more reliable, the task becomes much simpler. In that case, one might wonder why not simply optimize the prompt or training objective to explicitly instruct the model to follow the context. The long-context scenario might make this harder, but benchmarks used in the paper (e.g, ConflictQA) involve short passages that are far below the model’s context limit.

2. **Missing benchmarks**. There exist several datasets [1, 2, 3] that explicitly evaluate knowledge conflict resolution, but the paper only reports results on ConflictQA, while the rest are knowledge editing benchmarks, which only partially capture the intended problem.

3. **Missing baselines.** Numerous prior works, both prompting-based and training-based, directly tackle knowledge conflict resolution, yet none are included as baselines [3, 4, 5, 6]. The authors should either compare with or at least discuss why these methods were omitted.

[1] “ClashEval: Quantifying the tug-of-war between an LLM’s internal prior and external evidence”, NeurIPS 2025  \
[2] “FaithEval: Can Your Language Model Stay Faithful to Context, Even If 'The Moon is Made of Marshmallows'”, ICLR 2025 \
[3] “To Trust or Not to Trust? Enhancing Large Language Models' Situated Faithfulness to External Contexts”, ICLR 2025 \
[4] “KnowPO: Knowledge-aware Preference Optimization for Controllable Knowledge Selection in Retrieval-Augmented Language Models”, AAAI 2025 \
[5] “FaithfulRAG: Fact-Level Conflict Modeling for Context-Faithful Retrieval-Augmented Generation”, ACL 2025 \
[6] “Trusting Your Evidence: Hallucinate Less with Context-aware Decoding”, NAACL 2024

**Questions:**

See weakness.

---

> ### Author Response · Authors · 2025-11-14
>
> We thanks for your feedback. For some misunderstanding of our paper, we cover them in the **`Global Response`**.
>
> 1. Re:*Problem Scope and "Determining Trust"*
> The reviewer claims our work "reduces the problem to enforcing context faithfulness" rather than "truly deciding" which source to trust.
> This is precisely the same `"Control vs. Decision"` misinterpretation. Please refer to **`Global Response`**.
> The reviewer asks, *"why not simply optimize the prompt or training objective"* This reveals a misunderstanding of the problem's intractability:
>   - `"Optimizing the prompt"`: This is a circular fallacy, as we detail in the `Global Response`. **`A model that already fails to follow the provided context cannot be expected to reliably follow another contextual instruction (i.e., the prompt).`** Our IKE/"Vanilla Res" baseline (Table 1-3) is this "prompting" baseline, and its failure is what proves a mechanistic solution is necessary.
>   - `"Optimizing the training objective"`: We did compare against a comprehensive suite of training-based methods. Table 1 is almost entirely dedicated to them:
>     - Fine-tuning methods (FT-C, LORA)
>     - Parameter-editing methods (ROME, R-ROME, MEMIT, AlphaEdit)
>     - Memory-based methods (GRACE, WISE)
> As Table 1 shows, these methods are not a viable alternative. They are catastrophically unstable, failing to resolve the conflict (e.g., Efficacy near 0.0 for MEMIT) while simultaneously destroying the model's general knowledge (e.g., Locality near 0.0). In our industrial use cases, where stability is non-negotiable, training-free mechanism is far more practical.
>
> 2. Re:*comparison under long context scenario*
> The reviewer dismisses our benchmarks as "short passages." This was an intentional experimental design choice to isolate the mechanism of conflict.
> By using short passages, we eliminate **`confounding factors`** like "lost-in-the-middle" retrieval failures and force the model into a direct confrontation between the two knowledge streams. This proves that the failure is not one of retrieval, but of reconciliation.
>
> **This conflict problem is worse, not better, in long-context scenarios, as the contextual signal becomes even more diluted and easier for the model to ignore.** `Our method, by mechanistically amplifying the Attention stream` ($\mathbf{A}^{(l)}$), `is ideally suited to solve this.`
>
> 3. Re:*other benchmarks*
> First, we chose ZsRE, CounterFact and ConflictQA because they are the strictest test of this problem: they represent a complete conflict where the model's internal fact must be overwritten. The low "Original" scores in Table 1-3 confirm the high conflict ratio, validating our choice.
> Second, per the reviewer's suggestion, we are running additional experiments, and will present the results in the following discussion.
> Finally, we have added a discussion of the other works [3,4,5] in the rebuttal version.

---

### Official Review · Reviewer_CNSf · 2025-11-01

**Soundness:** 2
**Presentation:** 3
**Contribution:** 3
**Rating:** 2
**Confidence:** 4

**Summary:**

This paper proposes AdaRes, a parameter-free, inference-time mechanism to address knowledge conflicts between an LLM's internal (parametric) knowledge and external (contextual) information. The method is inspired by interpretability findings that identify the Attention module as the context aggregator and the FFN as the knowledge store. AdaRes calculates instance-specific "trust scores" for each source ($\alpha^{(l)}$ for context, $\beta^{(l)}$ for parametric) and uses them to reweight the respective contributions within the residual pathway. The method reports empirical results on knowledge editing and conflict-aware QA benchmarks.

**Strengths:**

The paper introduces a novel, training-free mechanism to address the critical problem of knowledge conflicts in LLMs, which is highly relevant for improving the reliability of RAG. The core idea is well-motivated by mechanistic interpretability findings, specifically the distinct roles of the Attention (context) and FFN (parametric) modules.

**Weaknesses:**

- The method's design is heavily "context-first" (as seen in the focus on Scenario #4) and seems to require a priori knowledge that the context should be trusted. This is a significant limitation that is not clearly acknowledged. A simple but crucial baseline is missing: prompt engineering. The results in Table 3 show low performance for the "Original" baseline even on instruction-tuned (it) models, suggesting that a simple, well-crafted prompt to "follow the context" was not explored as a point of comparison. The "IKE" results hint on a positive role of the instruction for Qwen 2.5. The actual prompts used are not disclosed, which harms reproducibility.
- The paper's description of the methodology is lacking. Important details about the FFN probing mechanism (for $\beta^{(l)}$ estimation) are relegated to the appendix. Furthermore, a "Top-n" selection is depicted in Figure 2 but never explained in the text, and it appears to be a hyperparameter. This contradicts the claim that the set of target layers $\mathcal{H}$ is the "sole hyperparameter" (line 229). The description of the context trust estimation ($\alpha^{(l)}$) is also unnecessarily convoluted.
- The submission is missing highly relevant citations in its related work (Section 2.2) regarding dual-response or fusion strategies. For example, [Huang et al. (ICLR 2025)](https://openreview.net/forum?id=K2jOacHUlO) addresses the identical problem of "dynamically calibrating trust" to "resolve knowledge conflicts" and should be discussed.
- The claim of "negligible runtime cost" is questionable. Algorithm 1 and the three-stream design imply that at least one additional, full forward pass is required for the probes. The significant increase in inference time in Figure 6 demonstrates this cost, while the paper downplays it.

**Questions:**

- How is AdaRes intended to operate when it is not known a priori whether the context or the parametric knowledge is correct? What happens if it is applied in Scenario #2 (correct model, wrong context)?
- Can you please disclose the prompts used for the "IKE" baseline? A comparison against a strong, instruction-based prompt (e.g., "Follow the context provided exactly") seems like a critical and missing baseline.
- Please clarify the "Top-n" selection from Figure 2. Is this a hyperparameter, and how was it set? This contradicts the claim that the layer set $\mathcal{H}$ is the sole hyperparameter.

### Minor Issues

- Table results (e.g., Table 1) would be more readable as percentages.
- Tables 2 and 3 are difficult to parse; adding a separate header row for model labels (e.g., Phi3, Gemma3) would improve clarity.
- There are minor typos (e.g., "an" -> "a" in line 248).

---

> ### Author Response · Authors · 2025-11-14
>
> Thanks for your feedback, we will address each point. **For the most critical misunderstandings regarding the "context-first" design and the prompting baseline, we cover them in our Global Response.**
>
> 1. Re:*the baseline of "Original"*
> The reviewer has misunderstood the purpose of the "Original" results in Table 1, 2 and 3. This is not a failed baseline.
> As explained in the submission (`Lines 254-255`) , "Original" measures the model's accuracy without any context provided. The extremely low score (e.g., 0.87% Efficacy on CounterFact for Llama3) is the desired result. It proves that for over 99% of the items, the model's internal knowledge directly conflicts with the fact we intend to provide in the context, confirming the dataset's suitability for this conflict-resolution task.
>
> 2. Re:*$\beta^{(l)}$ is explained in Appendix* and *topn*
> - $\beta^{(l)}$ Estimation: Due to the 9-page limit and the highly analogous nature of the calculations for $\alpha^{(l)}$ and $\beta^{(l)}$, we placed the detailed derivation in Appendix D to improve the main paper's readability.
> - "Top-n": It is a schematic representation of the averaging process, not a separate hyperparameter. As detailed in the text, the trust score is the Mean of token scores, meaning the operation is performed over all N query tokens. Thus, n=N, and there is no "Top-n" selection hyperparameter. Our claim that H (the layer set) is the "sole hyperparameter" is correct. `We have revised this in the rebuttal version to prevent misunderstandings.`
>
> 3. Re:*$\alpha^{(l)}$ computation*
> We respectfully disagree that *the description of $\alpha^{(l)}$ is "unnecessarily convoluted"*. The reviewer's narrow focus on this specific calculation reveals a superficial understanding of our multi-aspect contribution. The critique fixates on one component while missing the foundational mechanism it serves.
> Our innovation is best understood as a hierarchy:
> - **Core Mechanism Innovation**: Our primary contribution is the introduction of a "Lambda Function" (Eq.3) as a lightweight **`surrogate`** for the vanilla residual pathway. This mechanism itself is the *foundational discovery* that enables training-free control over the information flow.
> - **Implementation**: The trust estimation in Eq.4 is a subsequent contribution, representing one possible way to generate the control signals. This is our "training-free trust estimation".
> - **Trust Utilization**: How to use these signals (i.e., the asymmetric scaling in Eq.5) is another design choice, tailored for a specific goal.
>
> This philosophy is **`analogous to the "Lambda Function" concept in Python programming (which inspired its name)`**. Our Eq.3 provides the flexible framework (the lambda: ... construct), while Eqs.4 and 5 provide a specific, concrete instantiation (e.g., lambda x, y: (1 + x/(x+y))).
>
> Additionally, we focused the specific instantiation on trusting context, as it is a critical failure mode in RAG that we have observed extensively in our practical applications. This focus does not limit the generality of the core $\Lambda$ mechanism, which can be easily re-instantiated to serve other goals, such as trusting parametric knowledge (`Appendix F`).
>
> Thus, our contribution is not just "a convoluted way to calculate $\alpha$"; it is a novel **`lambda surrogate`** of vanilla residual in Transformer. **`We hope this core design philosophy is now clear, and we welcome further discussion.`**
>
> 4. Re:*negligible cost*
> We must reframe this "cost." The reviewer's comparison to baseline inference is flawed. The correct comparison is against parameter-editing alternatives (e.g., ROME), which require costly, persistent modifications.
> Quantitatively, ROME takes ~29 seconds per edit; our training-free AdaRes adds only ~0.25 seconds per pass on Llama3-8B (over 100x faster).
>
> Furthermore, reliability is paramount. As shown in Table 1, those editing methods are catastrophically unstable (e.g., Locality scores near 0.0), destroying general knowledge and rendering them impractical.
> In our industrial use cases, where stability is non-negotiable, training-free mechanism is far more practical. The modest cost in Figure 6 is a highly acceptable trade-off to avoid this unreliability.
>
> We have revised the term to "modest" in the rebuttal version, **but we maintain that this cost is, in all practical respects, negligible compared to the alternatives.**
>
> 5. Re:*discussion of ICLR 2025*
> We discussed it in the rebuttal version.
>
> 6. Re:*minor issues*
> For the metrics in Table 1, we adopted directly from the standard knowledge-editing evaluation literature to ensure a direct comparison. For the table formatting, this was done solely to save space. We have already provided de-merged tables in the rebuttal version, of course, use this formatting in the camera-ready version (if possible), where an additional page is permitted.

---

### Author Response · Authors · 2025-11-14
**Global Response (to all Reviewers): Clarification of "Decision vs. Control" and Prompting Baselines, etc.**

We thank all reviewers for their feedback. We must identify several recurring, fundamental misunderstandings of our work's core contribution.

---
All reviewers critique AdaRes based on a central misinterpretation: assuming it is a decision-making system that judges "correctness". **It is not**. Our method is a `control mechanism to govern the model's reliance`.

---
More importantly, **`Are the contextual/internal knowledge correct?` and `How to control model behavior?` are two fundamentally distinct, orthogonal questions.** **`Do you believe they are the same problem? Must these two issues necessarily influence each other?`** The reviewers' core criticism ("what if the context is wrong?") repeatedly conflates these two issues. We must disentangle them:
- The Problem of **`Decision`**: Which knowledge source is correct?
- The Problem of **`Control`**: How can we ensure the model trusts the source we command it to trust?

AdaRes is the first *parameter-free* and *prompt-free* solution to the Problem of **`Control`**.

---
From this decomposition, two aspects worthy of discussion are derived, which we must further clarify:
1. **`Control, not "correctness", dictates model behavior.`** **The reviewers' focus on "correctness" is a red herring**: **Does a model must use the internal "correct" knowledge when the contextual information is "incorrect" (`just because context is "wrong"`)?** In our business scenarios, a reliable system must depend on the use case (**not on a simplistic notion of "correctness"**):
  - **Use Cases (Trust Context even if it is "wrong")**: Consider the reviewers' own "what if context is wrong?". For a creative writing task where the prompt is "*In a world where the sky is green and the moon is made of cheese...*", the model's "correct" internal knowledge (*sky is blue*) must be suppressed in favor of the "factually incorrect" context. Similarly, considering another case, a customer support bot answering for a specific user's policy document (potentially flawed) must be controlled to follow that context, not its general parametric knowledge.

In these scenarios, **context appears "incorrect," yet the application demands we follow it.** This illustrates `the correctness of internal and external knowledge` is not related to `the control of model behavior`. **The only thing that matters is having a reliable mechanism to control the knowledge flow as the application demands**. Our method provides this control.

2. Moreover, **correctness is relative, which makes "Control" paramount.** In industrial applications (e.g., at Reality Labs), correctness is task-defined, not absolute. Our models must process time-sensitive, domain-specific information, such as a new, internal-only API specification. This new context is "correct" for the task, even though it directly conflicts with the model's "correct" public knowledge of an older, deprecated API. The model cannot be used to simply "decide"; it must be controlled to adhere to the provided context.

---
The reviewers propose using a prompt (e.g., "you must follow the context") to control the model. We must point out `the fundamental circularity in this suggestion`. **This instruction is itself delivered as part of the context**. **`If the model already fails to reliably follow the provided context (the very problem we address), why would we expect it to suddenly follow another contextual instruction commanding it to do so?`** This is precisely the failure of context-based control that necessitates a mechanistic solution like AdaRes.

Furthermore, **we did empirically evaluate this suggested "prompting" strategy**, using the exact prompt: "*Answer the question based on the given information*" (see Line 96, 144, 253, 286 in `adares_main.py`).
  - In the knowledge editing domain (Table 1 & 3), this is the established IKE baseline. As our paper shows, its performance is quite low (e.g., Llama3-Instruct Efficacy 0.56% on CounterFact).
  - On ConflictQA (Table 2), this prompt-based baseline is labeled "Vanilla Res".

In all cases, our mechanistic AdaRes dramatically outperforms these prompt-only methods, proving that simple prompting is not a viable solution for strong knowledge conflicts.

---
Finally, our work is general, not just "context-first". We focused experiments on Scenario #4 **`because it is a critical failure mode in RAG`**. However, AdaRes is fully general. As explicitly detailed in Appendix F&G, the asymmetric Lambda function can be inverted to force the model to trust its internal knowledge, by protecting and amplifying the FFN stream instead.

---
Therefore, **our core contribution is not another "decider."** It is a **foundational control mechanism** that enables, for the first time, explicit, lightweight, and dynamic calibration of the knowledge flow within a Transformer. This is a novel discovery that, **as we have already observed in our own application scenarios**, holds significant practical value for deploying reliable LLMs.

---

### Note · Authors · 2025-12-31

I have read and agree with the venue's withdrawal policy on behalf of myself and my co-authors.